# Alpha-IoU: A Family of Power Intersection over Union Losses for Bounding Box Regression

**Jiabo He[1,3,∗], Sarah Erfani[1], Xingjun Ma[2,†], James Bailey[1], Ying Chi[3,†], Xian-Sheng Hua[3]**

[1]School of Computing and Information Systems, The University of Melbourne
[2]School of Computer Science, Fudan University
[3]DAMO Academy, Alibaba Group
{jiaboh@student., sarah.erfani@, baileyj@}unimelb.edu.au
danxjma@gmail.com, {xinyi.cy, xiansheng.hxs}@alibaba-inc.com

## Abstract

Bounding box (bbox) regression is a fundamental task in computer vision. So far, the most commonly used loss functions for bbox regression are the Intersection over Union (IoU) loss and its variants. In this paper, we generalize existing IoU-based losses to a new family of power IoU losses that have a power IoU term and an additional power regularization term with a single power parameter $\alpha$. We call this new family of losses the $\alpha$-IoU losses and analyze properties such as order preservingness and loss/gradient reweighting. Experiments on multiple object detection benchmarks and models demonstrate that $\alpha$-IoU losses, 1) can surpass existing IoU-based losses by a noticeable performance margin; 2) offer detectors more flexibility in achieving different levels of bbox regression accuracy by modulating $\alpha$; and 3) are more robust to small datasets and noisy bboxes.

## 1 Introduction

Bounding box (bbox) regression localizes an object in an image/video by predicting a bbox for the object, which is fundamental to object detection, localization, and tracking. For example, the most advanced object detectors often consist of a bbox regression branch and a classification branch with the bbox regression branch generating bboxes to localize objects for classification. In this work, we explore more effective loss functions for bbox regression in the context of object detection.

Whilst early works in object detection use $\ell_n$-norm losses [11] for bbox regression, recent works directly adopt the localization performance metric, i.e., Intersection over Union (IoU), as the localization loss [28, 39]. Compared with $\ell_n$-norm losses, the IoU loss is invariant to bbox scales, thus helping train better detectors. However, the IoU loss suffers from the gradient vanishing problem when the predicted bboxes are not overlapping with the ground truth, which tends to slow down convergence and result in inaccurate detectors. This has motivated the design of several improved IoU-based losses including Generalized IoU (GIoU), Distance-IoU (DIoU) and Complete IoU (CIoU). GIoU introduces a penalty term into the IoU loss to alleviate the gradient vanishing problem [32], while DIoU and CIoU consider the central point distance and aspect ratio between predicted bboxes and their ground truth in penalty terms [43].

In this paper, we present a new family of IoU losses obtained by applying power transformations to existing IoU-based losses. We first apply the Box-Cox transformation [2] to the IoU loss $\mathcal{L}_{\text{IoU}} = 1 - IoU$ and generalize it to a power IoU loss: $\mathcal{L}_{\alpha\text{-IoU}} = (1 - IoU^\alpha)/\alpha, \ \alpha > 0$, denoted as $\alpha$-IoU. We further simplify $\alpha$-IoU to $\mathcal{L}_{\alpha\text{-IoU}} = 1 - IoU^\alpha$ for $\alpha \nrightarrow 0$ and extend it to a more general form

35th Conference on Neural Information Processing Systems (NeurIPS 2021).

---

[∗]This research was done when Jiabo He interned with the DAMO Academy, Alibaba Group.
[†]Corresponding authors.

with an additional power regularization term (see equation (3)). This allows us to generalize existing IoU-based losses, including GIoU, DIoU, and CIoU, to a new family of power IoU losses (see equation (4)) for more accurate bbox regression as well as object detection.

We show that, relative to $\mathcal{L}_{\text{IoU}}$, $\mathcal{L}_{\alpha\text{-IoU}}$ with $\alpha > 1$ up-weights both the loss and gradient of high IoU objects, leading to improved bbox regression accuracy. When $0 < \alpha < 1$, it down-weights high IoU objects which we find hurts regression accuracy. The power parameter $\alpha$ can serve as a knob to adapt $\alpha$-IoU losses to meeting different levels of bbox regression accuracy (precision measured under different IoU thresholds), with $\alpha > 1$ for high regression accuracy (i.e., high IoU thresholds) by focusing more on those high IoU objects. We also empirically show that $\alpha$ is not overly sensitive to different models or datasets, with $\alpha = 3$ performing consistently well in most cases. The family of $\alpha$-IoU losses can be easily applied for improving state-of-the-art detectors under both clean and noisy bbox settings without introducing additional parameters to these models (making any modifications to training algorithms), nor increasing their training/inference time.

In summary, our main contributions are as follows:

- We propose a new family of power IoU losses called $\alpha$-IoU for accurate bbox regression and object detection. $\alpha$-IoU presents a unified power generalization of existing IoU-based losses.

- We analyze a set of properties of $\alpha$-IoU, including order preservingness and loss/gradient reweighting, to show that a proper choice of $\alpha$ (i.e., $\alpha > 1$) can help improve bbox regression accuracy by adaptively up-weighting the loss and gradient of high IoU objects.

- We empirically show, on multiple benchmark object detection datasets and models, that $\alpha$-IoU losses can consistently outperform existing IoU-based losses and provide more robustness for small datasets and noisy bboxes.

## 2   Related Work

**Object Detection Models.** There exist two mainstream types of detection models: anchor-based and anchor-free detectors. Anchor-based detectors can be further divided into two-stage and one-stage models. Two-stage anchor-based detectors (e.g., R-CNN series [11, 31, 14, 3], HTC [5], and TSD [33]) are firstly proposed in object detection tasks, which are composed of region proposal networks (RPNs) and classifiers. RPNs generate a large number of foreground and background region proposals, followed by networks to classify objects in the proposals. Towards real-time object detection, one-stage anchor-based detectors (e.g., YOLO series [29, 30, 1], RetinaNet [21], and SSD [24]) are developed to predict bboxes and categories at the same time, thus no longer need RPNs. Anchor boxes with prior scales and aspect ratios should be defined before training anchor-based detectors. Techniques have been proposed to mitigate the sensitivity of these models to hand-picked anchor boxes, for example, attention-based fusion networks [31] and clustering algorithms [30]. These techniques learn prior anchors from the training set for every sliding window or grid cell.

Recently, anchor-free detectors such as CornerNet [16], CenterNet$_1$ [8], ExtremeNet [45], and CentripetalNet [7], have also been proposed to get rid of anchor priors. These models first predict locations of keypoints (corners, centroids, or extreme points), then group them into the same bboxes if they are geometrically aligned. There also exist other models that generate pixel-wise results. For example, CenterNet$_2$ estimates pixel-level categories of objects along with their sizes and offsets [44]. FCOS generates pixel-wise classification, centerness, and bbox (top, down, left, right) results using multi-head CNNs [34], followed by the Adaptive Training Sample Selection (ATSS) [40] as an improvement on automatically selecting positive and negative samples. In addition, transformers (e.g., DETR series [4, 46]) have also been developed for object detection without anchor generation or non-maximum suppression (NMS), achieving the performance on par with the above CNN-based detectors. In this work, we propose a new family of generalized IoU losses to improve the performance of these detectors without any architectural modifications, which is orthogonal to the above research.

**Bounding Box Regression Losses.** Anchor-based detectors regress offsets between ground-truth bboxes and their closest anchors, while anchor-free detectors predict keypoints of objects with some frameworks also generating the sizes of the bboxes. The predicted offsets or keypoints (w/ or w/o bbox sizes) are then mapped back to the pixel space for generating the bboxes. Localization losses usually compare the generated bboxes with their ground truth. Early works adopt $\ell_n$-norm losses [11] for bbox regression, which have been found sensitive to varying bbox scales. Recent works replace

them with the IoU loss and its variants such as BIoU, GIoU, DIoU and CIoU for bbox regression, as IoU is the metric for localization and it is scale-invariant [28, 39]. The Bounded IoU (BIoU) loss maximizes the IoU overlap between the region of interest (RoI) and the ground truth based on a set of IoU upper bounds [35]. GIoU is proposed to address the problem of gradient vanishing on non-overlapping examples, which are examples having non-overlapping predicted bboxes with the ground truth (IoU is zero) [32]. DIoU and CIoU [43] losses further consider the overlapping area, central point distance, and aspect ratio in IoU and the regularization terms. These regularization terms can help improve the convergence speed as well as the final detection performance. There are also losses designed to *focus more on high IoU objects*, for example, the Rectified IoU (RIoU) loss [36], and the Focal and Efficient IoU (Focal-EIoU) loss [41]. These loss functions increase gradients of those examples that are in high bbox regression accuracy. However, RIoU and Focal-EIoU are neither concise nor generalized compared with other IoU-based losses. In this paper, we apply a power transformation to generalize the above vanilla IoU loss and regularized IoU-based losses for both their IoU and regularization terms. The new family of generalized losses improve bbox regression accuracy by adaptively reweighting the loss and gradient of high and low IoU objects.

There are also works on AutoML-based loss function search for computer vision tasks [23, 18, 17]. Despite their advantage in saving human efforts, these methods are very expensive in searching qualified loss functions (e.g., days of searching time on multiple GPUs), and probably with limited performance improvement based on existing losses [23]. We will empirically compare with one of these losses in our experiments.

## 3 $\alpha$-IoU Losses for Bounding Box Regression

### 3.1 Preliminaries

We study the problem of bbox regression in object detection. Let $\boldsymbol{X} \in \mathbb{R}^{d_x}$ be the input space and $\boldsymbol{Y} \in \mathbb{R}^{d_y}$ be the annotation space, with $d_x$ and $d_y$ denoting the input and annotation dimensions, respectively. Given a dataset $D = \{(\boldsymbol{x}_i, \boldsymbol{y}_i)\}_{i=1}^{n}$ of $n$ training examples with each $(\boldsymbol{x}_i, \boldsymbol{y}_i) \in (\boldsymbol{X} \times \boldsymbol{Y})$, the task is to learn a function $f$ (represented by a detector network) that maps the input space to the annotation space $f : \boldsymbol{X} \to \boldsymbol{Y}$. In object detection, each $\boldsymbol{y}_i = (c_{i,k}, B_{i,k})_{k=1}^{m_i}$, where $m_i$ is the total number of objects in $\boldsymbol{x}_i$, $c_{i,k}$ is the category of the $k^{th}$ object in $\boldsymbol{x}_i$ and $B_{i,k}$ is its bbox.

The bbox regression performance is measured by the Intersection over Union (IoU) metric between the predicted bbox $B$ and the ground truth $B^{gt}$: $IoU = |B \cap B^{gt}|/|B \cup B^{gt}|$. Positive examples (both true and false positives) are determined from the set of predictions according to an IoU threshold, based on which the Average Precision (AP) over all categories of objects can be calculated. E.g., $\text{AP}_{50}$ measures the AP of objects localized by bboxes with an IoU that is above the threshold 0.5. The final performance of a detector is commonly evaluated by the mean Average Precision (mAP) across multiple IoU thresholds. For instance, the popular metric $\text{mAP}_{50:95}$ measures the mAP of examples across the set of IoU thresholds ranging from 0.5 to 0.95 with a stride of 0.05.

### 3.2 $\alpha$-IoU Losses

The vanilla IoU loss is defined as $\mathcal{L}_{\text{IoU}} = 1 - IoU$. We first apply the Box-Cox transformation[3] [2] and generalize the IoU loss to an $\alpha$-IoU loss:

$$\mathcal{L}_{\alpha\text{-IoU}} = \frac{1 - IoU^{\alpha}}{\alpha}, \alpha > 0. \tag{1}$$

By modulating the parameter $\alpha$ in $\alpha$-IoU, one can derive most of the IoU terms in existing losses, e.g., $\log(IoU)$, $IoU$ and $IoU^2$. When $\alpha \to 0$, we obtain $\lim_{\alpha \to 0} \mathcal{L}_{\alpha\text{-IoU}} = -\log(IoU) = \mathcal{L}_{\log(IoU)}$ [39] (see the proof in Appendix A). We recover the IoU loss with $\alpha = 1$: $\mathcal{L}_{1\text{-IoU}} = 1 - IoU = \mathcal{L}_{\text{IoU}}$. And $\mathcal{L}_{2\text{-IoU}} = \frac{1}{2}(1 - IoU^2) = \frac{1}{2}\mathcal{L}_{\text{IoU}^2}$, when $\alpha = 2$. We can also extend the above $\alpha$-IoU formula to loss functions with multiple IoU terms (e.g. RIoU [36]) by using multiple $\alpha$ values.

---

[3]The Box-Cox transformation has also been applied for generalizing the Cross Entropy (CE) loss into the Generalized Cross Entropy (GCE) loss [42]. Such a generalization unifies the Mean Absolute Error (MAE) and the CE loss, increasing the robustness to noisy labels for classification tasks. However, GCE may slow down convergence and lead to degraded performance [37, 25]. In contrast, our $\alpha$-IoU generalization is for bbox regression and can improve the localization performance on both clean and noisy bbox datasets.

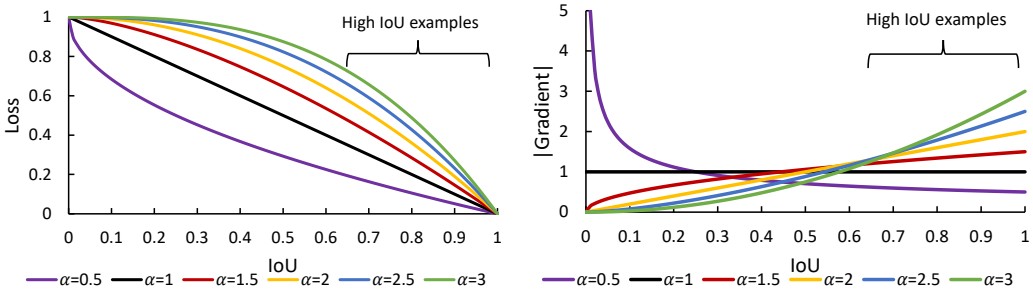

Figure 1: Correlation between IoU and $\mathcal{L}_{\alpha\text{-IoU}} = 1 - IoU^\alpha$ (left) and its absolute gradient $|\nabla_{\text{IoU}}\mathcal{L}_{\alpha\text{-IoU}}|$ (right) with different $\alpha \in [0.5, 3]$. According to both plots, $\mathcal{L}_{\alpha\text{-IoU}}$ reweights all objects adaptively and distinctively for $0 < \alpha < 1$ vs. $\alpha > 1$ ($\alpha = 1$ marks the IoU loss).

We simplify the above $\alpha$-IoU formula for $\alpha > 0$ and $\alpha \nrightarrow 0$, as in this case, the denominator $\alpha$ in equation (1) is just a positive constant in the objective. This gives us two cases of the $\alpha$-IoU loss for $\alpha \to 0$ and $\alpha \nrightarrow 0$, respectively:

$$\mathcal{L}_{\alpha\text{-IoU}} = \begin{cases} -\log(IoU), & \alpha \to 0, \\ 1 - IoU^\alpha, & \alpha \nrightarrow 0. \end{cases} \tag{2}$$

Here, we are more interested in the case $\alpha \nrightarrow 0$ as most state-of-the-art IoU-based losses have an $\alpha \geq 1$. We then extend the above $\alpha$-IoU loss for $\alpha \nrightarrow 0$ to a more general form by introducing a power penalty/regularization term into the formula:

$$\mathcal{L}_{\alpha\text{-IoU}} = 1 - IoU^{\alpha_1} + \mathcal{P}^{\alpha_2}(B, B^{gt}), \tag{3}$$

where $\alpha_1 > 0$, $\alpha_2 > 0$, and $\mathcal{P}^{\alpha_2}(B, B^{gt})$ denotes any penalty term computed based on $B$ and $B^{gt}$. This simple extension allows a straightforward generalization of existing IoU-based losses to their $\alpha$-IoU versions. In Appendix B.2.1, we empirically show that $\mathcal{L}_{\alpha\text{-IoU}}$ is not sensitive to $\alpha_2$. We thus maintain the power consistency between the IoU term and the penalty term and take $\alpha_1 = \alpha_2$ as a simple choice when training the detectors.

With the above $\alpha$-IoU formula, we can now generalize the commonly used IoU-based losses including $\mathcal{L}_{\text{IoU}}$, $\mathcal{L}_{\text{GIoU}}$, $\mathcal{L}_{\text{DIoU}}$, and $\mathcal{L}_{\text{CIoU}}$ using the same power parameter $\alpha$ for the IoU and penalty terms:

$$\mathcal{L}_{\text{IoU}} = 1 - IoU \implies \mathcal{L}_{\alpha\text{-IoU}} = 1 - IoU^\alpha,$$
$$\mathcal{L}_{\text{GIoU}} = 1 - IoU + \frac{|C \setminus (B \cup B^{gt})|}{|C|} \implies \mathcal{L}_{\alpha\text{-GIoU}} = 1 - IoU^\alpha + (\frac{|C \setminus (B \cup B^{gt})|}{|C|})^\alpha,$$
$$\mathcal{L}_{\text{DIoU}} = 1 - IoU + \frac{\rho^2(\boldsymbol{b}, \boldsymbol{b}^{gt})}{c^2} \implies \mathcal{L}_{\alpha\text{-DIoU}} = 1 - IoU^\alpha + \frac{\rho^{2\alpha}(\boldsymbol{b}, \boldsymbol{b}^{gt})}{c^{2\alpha}}, \tag{4}$$
$$\mathcal{L}_{\text{CIoU}} = 1 - IoU + \frac{\rho^2(\boldsymbol{b}, \boldsymbol{b}^{gt})}{c^2} + \beta v \implies \mathcal{L}_{\alpha\text{-CIoU}} = 1 - IoU^\alpha + \frac{\rho^{2\alpha}(\boldsymbol{b}, \boldsymbol{b}^{gt})}{c^{2\alpha}} + (\beta v)^\alpha,$$

where $C$ in $\mathcal{L}_{\text{GIoU}}$ denotes the smallest convex shape enclosing $B$ and $B^{gt}$; $\boldsymbol{b}$ and $\boldsymbol{b}^{gt}$ in $\mathcal{L}_{\text{DIoU}}$ denote central points of $B$ and $B^{gt}$ with $\rho(\cdot)$ being the Euclidean distance and $c$ being the diagonal length of the smallest enclosing box; and in $\mathcal{L}_{\text{CIoU}}$, $v = \frac{4}{\pi^2}(arctan\frac{w^{gt}}{h^{gt}} - arctan\frac{w}{h})^2$, $\beta = \frac{v}{(1-IoU)+v}$. They give us the family of power IoU losses for bbox regression with their original versions recovered at $\alpha = 1$. Note that the above $\alpha$-IoU generalization can be easily extended to more complex loss functions that have multiple IoU or penalty terms (e.g., $\mathcal{L}_{\alpha\text{-CIoU}}$). Next, we will analyze the properties of $\alpha$-IoU losses when $\alpha$ takes different values.

### 3.3   Properties of $\alpha$-IoU Losses

Here, we focus on the vanilla $\alpha$-IoU formula $\mathcal{L}_{\alpha\text{-IoU}} = 1 - IoU^\alpha$ to analyze its properties, as the penalty terms may affect these properties differently. Figure 1 illustrates the correlation between IoU and $\mathcal{L}_{\alpha\text{-IoU}}$ (left) and the magnitude of its gradient w.r.t. IoU, i.e., $|\nabla_{\text{IoU}}\mathcal{L}_{\alpha\text{-IoU}}|$ (right). One key observation is that the IoU loss (i.e., $\alpha = 1$) has a linear correlation with IoU and the gradient

is a constant, while $\mathcal{L}_{\alpha\text{-IoU}}$ reweights objects adaptively (according to their IoU values) following different reweighting schemes with $0 < \alpha < 1$ versus $\alpha > 1$.

The power transformation in $\mathcal{L}_{\alpha\text{-IoU}}$ preserves key properties of $\mathcal{L}_{\text{IoU}}$ as a performance metric, including non-negativity, identity of indiscernibles, symmetry, and triangle inequality [32]. Furthermore, we analyze the following important properties of $\mathcal{L}_{\alpha\text{-IoU}}$ with detailed derivations deferred to Appendix A. We first let $B_i$ and $B_j$ be two predicted bboxes by two different models $M_i$ and $M_j$ respectively, and $B_i$ and $B_j$ correspond to the same ground truth $B^{gt}$ with $IoU(B_i, B^{gt}) < IoU(B_j, B^{gt})$. Then we have the first property of $\mathcal{L}_{\alpha\text{-IoU}}$:

**Property 1** (Order Preservingness). *$\mathcal{L}_{\alpha\text{-IoU}}$ preserves the orders of both IoU and $\mathcal{L}_{IoU}$:* $IoU(B_i, B^{gt}) < IoU(B_j, B^{gt}) \iff \mathcal{L}_{IoU}(B_i, B^{gt}) > \mathcal{L}_{IoU}(B_j, B^{gt}) \iff \mathcal{L}_{\alpha\text{-IoU}}(B_i, B^{gt}) > \mathcal{L}_{\alpha\text{-IoU}}(B_j, B^{gt})$.

The above property indicates that both $\mathcal{L}_{\alpha\text{-IoU}}$ and $\mathcal{L}_{\text{IoU}}$ are monotonically decreasing functions w.r.t. $IoU$. As $\mathcal{L}_{\alpha\text{-IoU}}$ preserves the order of $\mathcal{L}_{\text{IoU}}$ strictly, it is guaranteed that $\arg\min_B \mathcal{L}_{\alpha\text{-IoU}}(B, B^{gt})$ is identical to $\arg\max_B IoU(B, B^{gt})$ and $\arg\min_B \mathcal{L}_{\text{IoU}}(B, B^{gt})$. In other words, the optimal solution $\arg\max_B IoU(B, B^{gt})$ can be obtained by minimizing either $\mathcal{L}_{\alpha\text{-IoU}}$ or $\mathcal{L}_{\text{IoU}}$. Following this, the adaptive relative loss reweighting scheme of $\mathcal{L}_{\alpha\text{-IoU}}$ can be characterized by the second property:

**Property 2** (Relative Loss Reweighting). *Compared with $\mathcal{L}_{IoU}$, $\mathcal{L}_{\alpha\text{-IoU}}$ adaptively reweights the relative loss of all objects by $w_{\mathcal{L}_r} = \mathcal{L}_{\alpha\text{-IoU}}/\mathcal{L}_{IoU} = 1 + (IoU - IoU^\alpha)/(1 - IoU)$, with $w_{\mathcal{L}_r}(IoU = 0) = 1$, and $\lim_{IoU \to 1} w_{\mathcal{L}_r} = \alpha$.*

The second property indicates that $\mathcal{L}_{\alpha\text{-IoU}}$ will adaptively down-weight and up-weight the relative loss of all objects according to their IoUs when $0 < \alpha < 1$ and $\alpha > 1$, respectively. We further note that, when $\alpha > 1$, the reweighting factor $w_{\mathcal{L}_r}$ increases monotonically with the increase of IoU ($w_{\mathcal{L}_r}$ grows from 1 to $\alpha$) while decreasing monotonically with the increase of IoU when $0 < \alpha < 1$ ($w_{\mathcal{L}_r}$ decays from 1 to $\alpha$). We will empirically show that the up-weighting scheme of $\mathcal{L}_{\alpha\text{-IoU}}$ with $\alpha > 1$ can help the model focus more on high IoU objects to improve both the localization (i.e., predict more high IoU objects) and detection (i.e., more accurate at high APs) performance[4]. Similarly, we can obtain the third property of adaptive relative gradient reweighting owned by $\mathcal{L}_{\alpha\text{-IoU}}$ as follows:

**Property 3** (Relative Gradient Reweighting). *Compared with $\mathcal{L}_{IoU}$, $\mathcal{L}_{\alpha\text{-IoU}}$ adaptively reweights the relative gradient of all objects by $w_{\nabla_r} = |\nabla_{IoU}\mathcal{L}_{\alpha\text{-IoU}}|/|\nabla_{IoU}\mathcal{L}_{IoU}| = \alpha IoU^{\alpha-1}$, with the turning point at $IoU = \alpha^{\frac{1}{1-\alpha}} \in (0, \frac{1}{e})$ when $0 < \alpha < 1$ and $IoU = \alpha^{\frac{1}{1-\alpha}} \in (\frac{1}{e}, 1)$ when $\alpha > 1$.*

When $\alpha > 1$, the above reweighting factor $w_{\nabla_r}$ increases monotonically with the increase of IoU, while decreasing monotonically with the increase of IoU when $0 < \alpha < 1$. This relative gradient reweighting scheme is also adaptive to IoU, with the turning point from up-weighting to down-weighting at $IoU = \alpha^{\frac{1}{1-\alpha}} \in (0, \frac{1}{e})$ when $0 < \alpha < 1$, and from down-weighting to up-weighting at $IoU = \alpha^{\frac{1}{1-\alpha}} \in (\frac{1}{e}, 1)$ when $\alpha > 1$. The gradient reweighting scheme is bounded by $w_{\nabla_r}(IoU = 1) = \alpha$, i.e., $0 \le w_{\nabla_r} \le \alpha$ when $\alpha > 1$, and $w_{\nabla_r} \ge \alpha$ when $0 < \alpha < 1$. This relative gradient reweighting scheme allows the model to learn objects with adaptive speeds (i.e., different gradients) according to their IoUs. Theoretically, when $\alpha = 2$, $|\nabla_{\text{IoU}}\mathcal{L}_{\alpha\text{-IoU}}| > |\nabla_{\text{IoU}}\mathcal{L}_{\text{IoU}}|$ for $IoU \in (0.5, 1]$, which accelerates the learning of all positive IoU objects at AP$_{50}$. However, we empirically show that $\alpha$-IoU losses with $\alpha = 3$ perform more competitively than those with $\alpha = 2$ in most cases. It is probable that $\alpha$-IoU losses with $\alpha = 3$ further up-weight the relative loss of objects with $IoU \in (0.5, 1]$, although $\alpha$-IoU losses with $\alpha = 2$ also beat existing baselines (see Figure 6). This property is both data-agnostic and model-agnostic, so we recommend $\alpha = 3$ or $\alpha \in [2, 3]$ in practical use for other datasets and models.

The above loss and gradient reweighting schemes can also be inferred from Figure 1, with detailed proofs in Appendix A. To summarize, $\mathcal{L}_{\alpha\text{-IoU}}$ trains better detectors than $\mathcal{L}_{\text{IoU}}$ for the following reasons. First, the same optimal IoU can be achieved by $\mathcal{L}_{\alpha\text{-IoU}}$ as that by $\mathcal{L}_{\text{IoU}}$ (**Property 1**). Second, $\mathcal{L}_{\alpha\text{-IoU}}$ with $\alpha > 1$ focuses more on high IoU objects by up-weighting their relative loss (**Property**

---

[4]The relative loss reweighting scheme of $\mathcal{L}_{\alpha\text{-IoU}}$ is reminiscent of the reweighting scheme of the focal loss $FL(\hat{p}) = -(1-\hat{p})^\gamma \log(\hat{p})$, $\gamma > 0$, which was proposed to encourage the learning of low confidence foreground objects in the presence of a large number of high confidence backgrounds [21]. In contrast, $\mathcal{L}_{\alpha\text{-IoU}}$ with $\alpha > 1$ focuses more on high IoU objects as the final performance is comprehensively measured by mAP$_{50:95}$, with low IoU objects (here $IoU < 0.5$) suppressed at evaluation. Note that focal loss is usually designed for the classification branch of object detectors while our $\alpha$-IoU is for the bbox regression branch.

**2**). Third, $\mathcal{L}_{\alpha\text{-IoU}}$ with $\alpha > 1$ helps detectors learn faster on high IoU objects (here $IoU \in (\alpha^{\frac{1}{1-\alpha}}, 1]$) through up-weighting their relative gradient (**Property 3**). In Appendix A, we also provide an analysis of the *absolute* loss and gradient reweighting properties (**Property 4** and **5**), showing the additions of $\alpha$-IoU to IoU. Specifically, when $\alpha > 1$, $\mathcal{L}_{\alpha\text{-IoU}}$ adds an absolute loss weight to $\mathcal{L}_{\text{IoU}}$ (i.e., $w_{\mathcal{L}_a} = \mathcal{L}_{\alpha\text{-IoU}} - \mathcal{L}_{\text{IoU}} = IoU - IoU^{\alpha} > 0$ for $IoU \in (0, 1)$), which creates more space for optimization on all levels of objects (**Property 4**). Likewise, $\mathcal{L}_{\alpha\text{-IoU}}$ puts an absolute gradient weight for high IoU objects (i.e., $w_{\nabla_a} = |\nabla_{\text{IoU}}\mathcal{L}_{\alpha\text{-IoU}}| - |\nabla_{\text{IoU}}\mathcal{L}_{\text{IoU}}| = \alpha IoU^{\alpha-1} - 1 > 0$ for $IoU \in (\alpha^{\frac{1}{1-\alpha}}, 1]$) such that the learning of high IoU objects is accelerated (**Property 5**). Both of the absolute and relative properties of $\mathcal{L}_{\alpha\text{-IoU}}$ are adaptive to the IoU values of the objects. Such reweighting schemes will provide more flexibility in achieving different levels of bbox regression accuracies (AP measured under different IoU thresholds).

**Learning Dynamics of $\mathcal{L}_{\alpha\text{-IoU}}$.** Training with $\mathcal{L}_{\alpha\text{-IoU}}$ is a dynamic process and should be interpreted based on both the absolute and relative properties. With $\alpha > 1$, easy examples will be learned first with increasing speed towards $IoU = 1$, while hard examples will be learned gradually and accelerated later on as their IoU improves. We will empirically show in Figure 3 that up-weighting the loss and gradient of high IoU objects can boost the training at the later stage. As a comparison, we will also show that $\alpha$-IoU losses with $0 < \alpha < 1$ tend to degrade the final performance in Section 4.4. Reducing the loss and gradient of high IoU objects ends up with more poorly localized objects.

## 4 Experiments

### 4.1 Datasets and Training Setup

We conduct all experiments on two popular benchmarks, i.e., PASCAL VOC [9] and MS COCO [22]. On the PASCAL VOC benchmark, we train all models on the trainval set 2007+2012 (containing $16, 551$ images from 20 categories) and evaluate them on the test set 2007 (containing $4, 952$ images) [9]. On the MS COCO benchmark, we train all models on the training set 2017 (containing 118K images from 80 categories) and evaluate them on the val set 2017 (containing 5K images) [22]. We train all state-of-the-art models with the original implementation released by the authors. Specifically, we follow the original implementation's training protocol with default parameters and the number of training epochs with different losses [31, 32, 43, 4]. Implementation details of all models are given in Appendix B.1. All experiments are run with NVIDIA V100 GPUs. Code is available at `https://github.com/Jacobi93/Alpha-IoU`.

### 4.2 Results and Analysis

We first validate the effectiveness of $\alpha$-IoU losses in training both anchor-based and anchor-free models on the two datasets. We choose YOLOv5s (i.e., YOLOv5 small) and YOLOv5x (i.e., YOLOv5 extra large) as one-stage anchor-based models, and DETR (ResNet-50) as an anchor-free model. Both $\alpha$-IoU losses (i.e., $\mathcal{L}_{\alpha\text{-IoU}}$ and $\mathcal{L}_{\alpha\text{-DIoU}}$) are generalized from existing baselines following equation (4). From Table 1, we can observe that $\alpha$-IoU losses surpass existing losses consistently across multiple models and datasets in terms of both mAP and mAP$_{75:95}$, especially at the high bbox regression accuracy mAP$_{75:95}$. The superiority of $\alpha$-IoU losses is more pronounced at high accuracy levels, which might reach more than $60\%$ relative improvement at AP$_{95}$. Interestingly, $\alpha$-IoU losses tend to help more of light models (e.g., YOLOv5s with 7.3M parameters and 17 GFLOPs) than heavy models (e.g., YOLOv5x with 87.7M parameters and 218.8 GFLOPs). This indicates that $\alpha$-IoU losses hold more advantage while training light models in computing-resource-limited scenarios, such as mobile devices, autonomous vehicles, and robots.

The consistent improvements on both PASCAL VOC and MS COCO demonstrate the stability of $\alpha$-IoU losses across different datasets. In addition, we also verify its robustness to extremely small training sets in Appendix B.2.2, where $\alpha$-IoU losses beat existing losses at various scales, i.e., from 4K ($25\%$ trainval set of PASCAL VOC 2007+2012) to 118K (the entire training set of MS COCO 2017) samples. It is possible that $\alpha$-IoU losses may not perform well if measured by a single low AP metric. For example, there may be less than $0.5\%$ performance drop at AP$_{50}$ when $\alpha = 3$, however, this is compensated by the significant boost at high APs. With some examples from the test set of PASCAL VOC 2007 (Figure 4) and the val set of MS COCO 2017 (Figure 5), we show that $\alpha$-IoU

Table 1: The performance of YOLOv5s, YOLOv5x and DETR models trained using different localization losses on PASCAL VOC and MS COCO benchmarks. Results are obtained on the test set of PASCAL VOC 2007 and the val set of MS COCO 2017. mAP denotes $mAP_{50:95}$; $mAP_{75:95}$ denotes the mean AP over $AP_{75}, AP_{80}, \cdots, AP_{95}$. "rela. improv." stands for the relative improvement. $\alpha = 3$ is used for all $\alpha$-IoU losses in all experiments.

| Method | Loss | PASCAL VOC | | | | | | MS COCO | | | | | |
|---|---|---|---|---|---|---|---|---|---|---|---|---|---|
| | | $AP_{50}$ | $AP_{75}$ | $AP_{85}$ | $AP_{95}$ | mAP | $mAP_{75:95}$ | $AP_{50}$ | $AP_{75}$ | $AP_{85}$ | $AP_{95}$ | mAP | $mAP_{75:95}$ |
| YOLOv5s | $\mathcal{L}_{IoU}$ | 78.81 | 58.04 | 35.07 | 2.34 | 52.74 | 32.45 | 55.51 | 38.59 | 23.58 | 2.07 | 36.29 | 21.82 |
| | $\mathcal{L}_{\alpha\text{-}IoU}$ | 78.62 | 58.78 | 38.16 | 3.64 | 53.61 | 34.46 | 55.25 | 39.69 | 25.85 | 3.35 | 37.01 | 23.66 |
| | rela. improv. | -0.24% | 1.27% | 8.81% | 55.56% | 1.65% | 6.21% | -0.47% | 2.85% | 9.63% | 61.84% | 1.98% | 8.43% |
| | $\mathcal{L}_{DIoU}$ | 78.19 | 57.77 | 34.89 | 2.36 | 52.30 | 32.17 | 55.67 | 39.01 | 23.56 | 2.03 | 36.36 | 21.95 |
| | $\mathcal{L}_{\alpha\text{-}DIoU}$ | 78.33 | 59.24 | 38.46 | 3.50 | 53.76 | 34.66 | 55.84 | 39.49 | 25.49 | 3.30 | 36.74 | 23.34 |
| | rela. improv. | 0.18% | 2.54% | 10.23% | 48.31% | 2.79% | 7.72% | 0.31% | 1.23% | 8.19% | 62.56% | 1.05% | 6.32% |
| YOLOv5x | $\mathcal{L}_{IoU}$ | 85.24 | 70.08 | 53.08 | 10.88 | 63.95 | 46.78 | 67.36 | 52.15 | 38.22 | 9.31 | 48.42 | 34.42 |
| | $\mathcal{L}_{\alpha\text{-}IoU}$ | 84.83 | 70.20 | 53.75 | 13.74 | 64.25 | 48.06 | 67.72 | 52.61 | 38.62 | 9.76 | 48.67 | 34.72 |
| | rela. improv. | -0.48% | 0.17% | 1.26% | 26.29% | 0.47% | 2.73% | 0.53% | 0.88% | 1.05% | 4.83% | 0.52% | 0.87% |
| | $\mathcal{L}_{DIoU}$ | 85.04 | 71.05 | 53.71 | 11.11 | 64.21 | 47.30 | 67.54 | 52.03 | 38.02 | 8.58 | 48.38 | 34.16 |
| | $\mathcal{L}_{\alpha\text{-}DIoU}$ | 84.90 | 71.34 | 54.23 | 13.85 | 64.49 | 48.40 | 67.42 | 52.65 | 39.28 | 10.29 | 48.81 | 35.42 |
| | rela. improv. | -0.16% | 0.41% | 0.97% | 24.66% | 0.44% | 2.32% | -0.18% | 1.19% | 3.31% | 19.93% | 0.89% | 3.68% |
| DETR | $\mathcal{L}_{IoU}$ | 76.50 | 53.85 | 29.54 | 1.62 | 49.78 | 28.82 | 59.38 | 41.67 | 26.13 | 3.52 | 39.23 | 24.37 |
| | $\mathcal{L}_{\alpha\text{-}IoU}$ | 76.22 | 55.03 | 32.30 | 2.28 | 51.12 | 31.08 | 59.61 | 42.65 | 28.57 | 5.09 | 40.18 | 26.44 |
| | rela. improv. | -0.37% | 2.19% | 9.34% | 40.74% | 2.69% | 7.84% | 0.39% | 2.35% | 9.34% | 44.60% | 2.42% | 8.49% |
| | $\mathcal{L}_{DIoU}$ | 76.26 | 54.09 | 29.23 | 1.56 | 49.91 | 28.68 | 59.28 | 41.62 | 26.09 | 3.54 | 39.25 | 24.48 |
| | $\mathcal{L}_{\alpha\text{-}DIoU}$ | 76.44 | 54.89 | 31.48 | 2.44 | 50.96 | 30.60 | 59.38 | 42.34 | 28.23 | 5.36 | 39.94 | 26.05 |
| | rela. improv. | 0.24% | 1.48% | 7.70% | 56.41% | 2.10% | 6.69% | 0.17% | 1.73% | 8.20% | 51.41% | 1.76% | 6.41% |

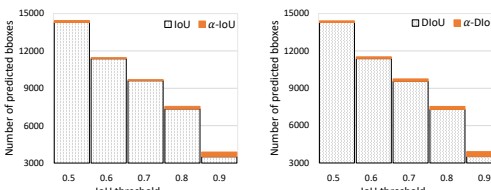

Figure 2: IoU distributions between predicted bboxes and their ground truth after NMS.

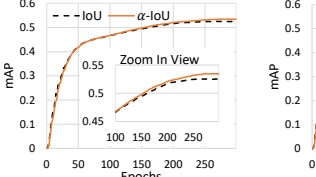 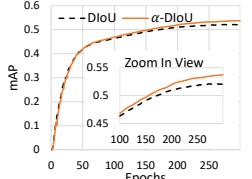

Figure 3: Validation mAPs ($mAP_{50:95}$) across 300 training epochs.

losses are able to localize objects more accurately than the baselines with more true positives and fewer false positives.

We further analyze the bbox regression accuracy by showing the IoU distributions between the predicted bboxes and their ground truth for YOLOv5s trained using different losses on PASCAL VOC. After NMS with the IoU threshold being $0.5$, we visualize the number of positively predicted bboxes under different IoU thresholds from $0.5$ to $0.9$ in Figure 2, showing that $\alpha$-IoU losses detect more positive objects than baseline losses across all IoU thresholds. Particularly, $\alpha$-IoU losses detect approximately $1\%$ more positive objects than the baselines when $IoU \geq 0.5$, and $11\%$ more high IoU objects when $IoU \geq 0.9$. This demonstrates that $\alpha$-IoU boosts both the precisions and recalls of detectors. $\alpha$-IoU is extremely advantageous in pushing low IoU objects to high IoU objects by up-weighting their loss, thus outperforming baseline losses significantly at the high accuracy level and contributing to the improvement of the final detection performance (Table 1).

Moreover, Figure 3 shows that $\alpha$-IoU losses are able to boost the late training stage (e.g., after 200 epochs) through up-weighting the gradient of high IoU objects, while almost having no negative impact on the early training stage (e.g., the first 100 epochs). When $\alpha > 1$, the relative gradient weight is $0 \leq w_{\nabla_r} < 1$ for $0 \leq IoU < \alpha^{\frac{1}{1-\alpha}}$, while $1 \leq w_{\nabla_r} \leq \alpha$ for $\alpha^{\frac{1}{1-\alpha}} \leq IoU \leq 1$, as analyzed in **Property 3** and illustrated in Figure 1 (right). This property helps tune down the gradients of low IoU objects at the early training stage, which has a smoothing effect (reduces the high variance in parameter update caused by hard examples) that helps stabilize the model training when gradients are large at the early stage. On the other hand, the gradient up-weighting is well-bounded by $w_{\nabla_r} \leq \alpha$, which makes up-weighting relatively safe for high IoU objects, as the original loss and gradient are small for these examples, so is the learning rate.

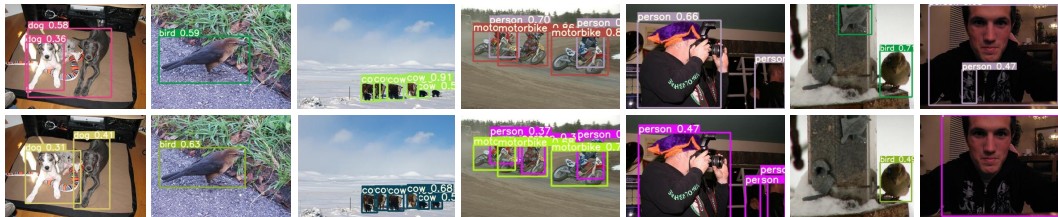

Figure 4: Example results on the test set of PASCAL VOC 2007 using YOLOv5s trained by $\mathcal{L}_{\text{IoU}}$ (top row) and $\mathcal{L}_{\alpha\text{-IoU}}$ with $\alpha = 3$ (bottom row). $\mathcal{L}_{\alpha\text{-IoU}}$ performs better than $\mathcal{L}_{\text{IoU}}$ because it can localize objects more accurately (image 1 and 2), thus can detect more true positive objects (image 3 to 5) and fewer false positive objects (image 6 and 7).

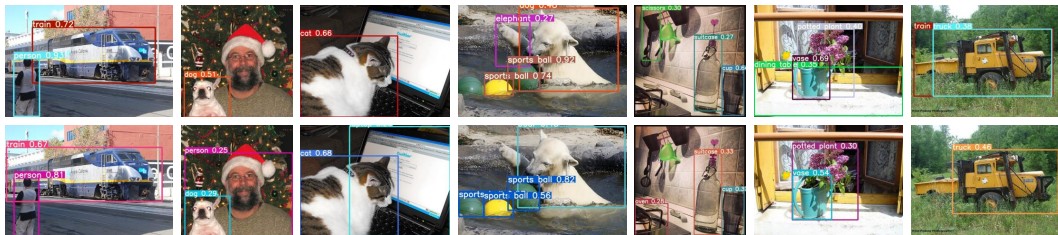

Figure 5: Example results on the val set of MS COCO 2017 using YOLOv5s trained by $\mathcal{L}_{\text{IoU}}$ (top row) and $\mathcal{L}_{\alpha\text{-IoU}}$ with $\alpha = 3$ (bottom row). $\mathcal{L}_{\alpha\text{-IoU}}$ performs better than $\mathcal{L}_{\text{IoU}}$ because it can localize objects more accurately (image 1), thus can detect more true positive objects (image 2 to 5) and fewer false positive objects (image 4 to 7). Note that $\mathcal{L}_{\alpha\text{-IoU}}$ detects both more true positive and fewer false positive objects in image 4 and 5 than $\mathcal{L}_{\text{IoU}}$.

We also conduct an experiment to compare our $\alpha$-IoU with a set of existing IoU-based losses in training a popular two-stage anchor-based model, Faster R-CNN (ResNet-50-FPN). In Table 2, results at the top are reproduced using the MMDetection toolbox [6] while those in the middle are reported results in the original papers [43, 41, 23]. Results at the bottom are obtained by replacing existing losses with their $\alpha$-IoU versions (i.e., improve based on top results using MMDetection). The results on MS COCO demonstrate that $\alpha$-IoU losses are quite competitive compared with existing baselines in terms of both mAP and mAP$_{75:95}$. Note that the Autoloss searches both the classification loss and the localization loss, thus taking a huge amount of searching time [23]. In contrast, $\alpha$-IoU losses only need an easy modification of the localization loss and win the Autoloss without causing any additional computational overhead.

## 4.3 Robustness to Noisy Bounding Boxes

It happens quite often that people annotate inaccurate bboxes in images/videos as the ground truth, even with computer-assisted annotation tools. However, there is little work on the robustness of localization losses to noisy bboxes, even though a number of methods have been proposed for robust learning with noisy labels, anchors, and bboxes [27, 26, 12, 10, 15, 37, 38, 25, 19, 20]. Here, we fill this gap by conducting a set of experiments to evaluate the robustness of different localization losses to noisy bboxes. We show that $\alpha$-IoU is more robust to noisy bboxes as they focus less on the low IoU objects, creating a suppression effect on the learning of the noisy bbox examples. Considering that open datasets like PASCAL VOC and MS COCO are carefully annotated, we synthesize a set of common noisy bboxes by perturbing normalized bboxes in the entire training set. The perturbations follow a uniform noise distribution in $[-\eta w, \eta w]$ at horizontal coordinates ($x$ and $w$) and $[-\eta h, \eta h]$ at vertical coordinates ($y$ and $h$), where $\eta$ is the noise rate [20]. We then constrain all the noisy bboxes by the following boundary conditions:

$$0 < w < 1, \ \ 0 < h < 1, \ \ \frac{1}{2}w \le x \le 1 - \frac{1}{2}w, \ \ \frac{1}{2}h \le y \le 1 - \frac{1}{2}h. \qquad (5)$$

Table 2: The performance of Faster R-CNN (ResNet-50-FPN) with $1\times$ schedule and single scale training on MS COCO using different localization losses. Results are obtained on the val set of MS COCO 2017. mAP denotes $mAP_{50:95}$; $mAP_{75:95}$ denotes the mean AP over $AP_{75}, AP_{80}, \cdots, AP_{95}$. $AP_s$, $AP_m$, and $AP_l$ denote the AP for small, medium, and large objects, respectively. $^\dagger$ marks the reproduced results from the MMDetection toolbox [6], while $^*$ marks the results in the original papers. "–" represents the missing results in papers. $\alpha = 3$ is used for all $\alpha$-IoU losses in all experiments. The top two best results in every column are **boldfaced**.

| Loss | $AP_{50}$ | $AP_{75}$ | $AP_{80}$ | $AP_{85}$ | $AP_{90}$ | $AP_{95}$ | **mAP** | **$mAP_{75:95}$** | $AP_s$ | $AP_m$ | $AP_l$ |
|---|---|---|---|---|---|---|---|---|---|---|---|
| $^\dagger\ell_1$ | 58.13 | 40.45 | 33.56 | 23.39 | 11.09 | 1.24 | 37.37 | 21.95 | 21.20 | 40.96 | 48.13 |
| $^\dagger\mathcal{L}_{IoU}$ | 58.12 | 41.23 | 34.03 | 24.43 | 12.42 | 1.61 | 37.88 | 22.74 | 21.61 | 41.63 | 49.11 |
| $^\dagger\mathcal{L}_{GIoU}$ | 58.18 | 41.00 | 33.52 | 24.13 | 11.97 | 1.51 | 37.62 | 22.43 | 21.49 | 41.07 | 48.90 |
| $^\dagger\mathcal{L}_{BIoU}$ | 58.05 | 40.57 | 33.54 | 23.85 | 11.10 | 1.19 | 37.43 | 22.05 | 21.57 | 41.00 | 48.17 |
| $^*\mathcal{L}_{IoU}$ | – | 40.79 | – | – | – | – | 37.93 | – | 21.58 | 40.82 | 50.14 |
| $^*\mathcal{L}_{GIoU}$ | – | 41.11 | – | – | – | – | 38.02 | – | 21.45 | 41.06 | 50.21 |
| $^*\mathcal{L}_{DIoU}$ | – | 41.11 | – | – | – | – | 38.09 | – | 21.66 | 41.18 | 50.32 |
| $^*\mathcal{L}_{CIoU}$ | – | 41.96 | – | – | – | – | 38.65 | – | 21.32 | 41.83 | **51.51** |
| $^*\mathcal{L}_{Focal-EIoU}$ | **59.10** | **42.40** | – | – | – | – | 38.90 | – | 21.20 | 41.10 | 50.20 |
| $^*$Autoloss | 58.60 | 41.80 | – | – | – | – | 38.50 | – | 22.00 | **42.20** | 50.20 |
| $\mathcal{L}_{\alpha\text{-}IoU}$ | 58.81 | 41.94 | 34.81 | **25.36** | **13.27** | 1.81 | 38.96 | 23.44 | **22.14** | 42.11 | 50.36 |
| $\mathcal{L}_{\alpha\text{-}GIoU}$ | 59.01 | 42.00 | **35.13** | 25.14 | 13.09 | **2.03** | 39.18 | 23.46 | 22.05 | **42.19** | 50.08 |
| $\mathcal{L}_{\alpha\text{-}DIoU}$ | **59.27** | **42.18** | **35.25** | **25.47** | **13.32** | 1.95 | **39.43** | **23.65** | 22.10 | 42.10 | **50.43** |
| $\mathcal{L}_{\alpha\text{-}CIoU}$ | 59.09 | 41.92 | 35.01 | 25.08 | 13.04 | **1.98** | **39.25** | 23.41 | 21.94 | 41.88 | 50.01 |

Table 3: The performance of YOLOv5s trained using different localization losses on simulated noisy trainval sets of PASCAL VOC 2007+2012 under noise rates $\eta = 0.1, 0.2$, and $0.3$. Results are obtained on the clean test set of PASCAL VOC 2007. mAP denotes $mAP_{50:95}$; $mAP_{75:95}$ denotes the mean AP over $AP_{75}, AP_{80}, \cdots, AP_{95}$. "rela. improv." stands for the relative improvement. $\alpha = 3$ is used for all $\alpha$-IoU losses in all experiments.

| Noise | Loss | $AP_{50}$ | $AP_{55}$ | $AP_{60}$ | $AP_{65}$ | $AP_{70}$ | $AP_{75}$ | $AP_{80}$ | $AP_{85}$ | $AP_{90}$ | $AP_{95}$ | **mAP** | **$mAP_{75:95}$** |
|---|---|---|---|---|---|---|---|---|---|---|---|---|---|
| 0.1 | $\mathcal{L}_{IoU}$ | 74.48 | 71.57 | 68.08 | 63.29 | 56.55 | 47.12 | 33.06 | 17.53 | 4.16 | 0.26 | 43.61 | 20.43 |
| | $\mathcal{L}_{\alpha\text{-}IoU}$ | 74.67 | 71.94 | 68.73 | 64.27 | 57.75 | 48.50 | 36.88 | 21.25 | 6.30 | 0.28 | 45.06 | 22.64 |
| | rela. improv. | 0.26% | 0.52% | 0.95% | 1.55% | 2.12% | 2.93% | 11.55% | 21.22% | 51.44% | 7.69% | 3.32% | 10.85% |
| | $\mathcal{L}_{DIoU}$ | 74.09 | 71.46 | 67.88 | 63.09 | 56.18 | 46.71 | 32.67 | 17.50 | 4.43 | 0.23 | 43.42 | 20.31 |
| | $\mathcal{L}_{\alpha\text{-}DIoU}$ | 74.38 | 71.95 | 68.10 | 63.52 | 57.18 | 48.47 | 35.90 | 20.89 | 6.37 | 0.33 | 44.71 | 22.39 |
| | rela. improv. | 0.39% | 0.69% | 0.32% | 0.68% | 1.78% | 3.77% | 9.89% | 19.37% | 43.79% | 43.48% | 2.97% | 10.26% |
| 0.2 | $\mathcal{L}_{IoU}$ | 67.82 | 63.93 | 58.22 | 50.11 | 39.31 | 26.33 | 13.51 | 4.55 | 0.66 | 0.05 | 32.45 | 9.02 |
| | $\mathcal{L}_{\alpha\text{-}IoU}$ | 68.20 | 64.21 | 58.77 | 51.59 | 40.66 | 29.20 | 16.11 | 6.06 | 1.31 | 0.10 | 33.62 | 10.56 |
| | rela. improv. | 0.56% | 0.44% | 0.94% | 2.95% | 3.43% | 10.90% | 19.25% | 33.19% | 98.48% | 100% | 3.61% | 17.03% |
| | $\mathcal{L}_{DIoU}$ | 67.39 | 62.94 | 57.29 | 49.25 | 39.40 | 27.13 | 13.78 | 4.52 | 0.68 | 0.02 | 32.24 | 9.23 |
| | $\mathcal{L}_{\alpha\text{-}DIoU}$ | 68.26 | 64.49 | 59.59 | 51.99 | 41.19 | 29.12 | 15.77 | 5.84 | 1.25 | 0.21 | 33.77 | 10.44 |
| | rela. improv. | 1.29% | 2.46% | 4.01% | 5.56% | 4.54% | 7.34% | 14.44% | 29.20% | 83.82% | 950% | 4.75% | 13.14% |
| 0.3 | $\mathcal{L}_{IoU}$ | 56.54 | 49.69 | 40.67 | 30.80 | 19.99 | 11.13 | 4.81 | 1.43 | 0.31 | 0.04 | 21.54 | 3.54 |
| | $\mathcal{L}_{\alpha\text{-}IoU}$ | 58.59 | 51.58 | 43.23 | 32.93 | 22.27 | 12.52 | 5.91 | 2.16 | 0.73 | 0.12 | 23.00 | 4.29 |
| | rela. improv. | 3.63% | 3.80% | 6.29% | 6.92% | 11.41% | 12.49% | 22.87% | 51.05% | 135% | 200% | 6.78% | 20.99% |
| | $\mathcal{L}_{DIoU}$ | 56.84 | 49.82 | 41.50 | 32.06 | 20.80 | 11.22 | 4.84 | 1.51 | 0.46 | 0.07 | 21.91 | 3.62 |
| | $\mathcal{L}_{\alpha\text{-}DIoU}$ | 58.45 | 51.94 | 43.9 | 33.78 | 22.57 | 12.89 | 6.34 | 2.42 | 0.65 | 0.16 | 23.31 | 4.49 |
| | rela. improv. | 2.83% | 4.26% | 5.78% | 5.36% | 8.51% | 14.88% | 30.99% | 60.26% | 41.30% | 129% | 6.39% | 24.09% |

We test $\eta = 0.1, 0.2, 0.3$ in our experiments, with the average IoU between the noisy bboxes and their clean versions dropping to $0.833$, $0.710$, and $0.613$, respectively. Examples of the synthesized noisy bboxes can be found in Appendix B.4. As shown in Table 3, $\alpha$-IoU improves the baseline losses (i.e., $\mathcal{L}_{IoU}$ and $\mathcal{L}_{DIoU}$) considerably in these noisy scenarios. We gain increasing relative improvements from $AP_{50}$ to $AP_{95}$, which accumulate to a more significant improvement in $mAP_{75:95}$. Note that $\alpha$-IoU losses also outperform the baselines at $AP_{50}$ across all noisy scenarios, which is not always the case when bboxes are clean (Table 1). Furthermore, $\alpha$-IoU losses are noticeably more robust against more severe noises. For instance, the relative improvement of $\mathcal{L}_{\alpha\text{-}DIoU}$ over $\mathcal{L}_{DIoU}$ increases from $2.97\%/10.26\%$ to $6.39\%/24.09\%$ according to mAP/$mAP_{75:95}$ when the noise rate $\eta$ rises from $0.1$ to $0.3$. These results confirm the advantage of $\alpha$-IoU losses in noisy bbox scenarios.

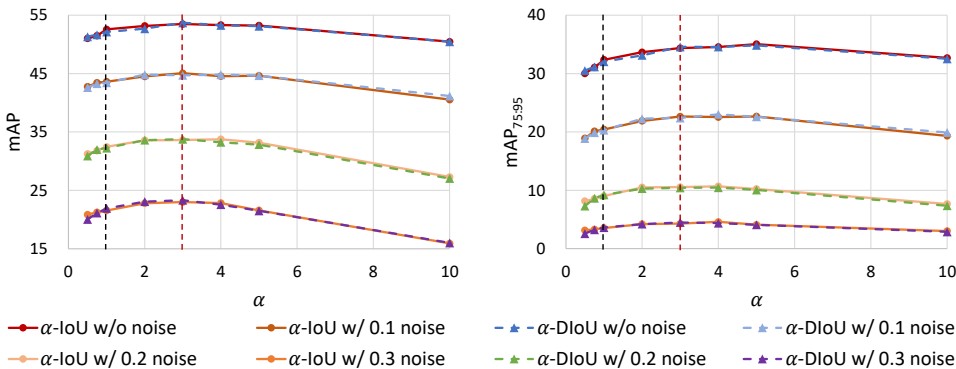

Figure 6: The performance of YOLOv5 models trained using $\alpha$-IoU with different $\alpha$ values and evaluated on the clean test set of PASCAL VOC 2007. Black dashed lines denote baselines (i.e., the family of $\alpha$-IoU with $\alpha = 1$) while red dashed lines denote the family of $\alpha$-IoU with $\alpha = 3$.

### 4.4 Sensitivity to power parameter $\alpha$

Here, we evaluate the performance of $\alpha$-IoU with varying $\alpha$ values ($\alpha \in [0.5, 5]$) via a set of experiments with $\mathcal{L}_{\alpha\text{-IoU}}$ and $\mathcal{L}_{\alpha\text{-DIoU}}$. The results are shown in Figure 6 for YOLOv5s on PASCAL VOC in both clean and various noisy bbox scenarios. It is evident that $\alpha$-IoU losses with $\alpha \in [2, 4]$ perform competitively well across all scenarios, with $\alpha = 3$ performing the best in most cases. When $\alpha > 3$, $\alpha$-IoU losses tend to perform worse on low APs than the baselines (i.e., $\alpha$-IoU with $\alpha = 1$), although the performance at high APs gains more improvement. We also test an extreme case with $\alpha = 10$, in which the performance drops by $5.61\%/10.92\%/23.88\%/31.82\%$ on average compared with $\alpha = 3$ under noise rates $\eta = 0/0.1/0.2/0.3$, respectively. More specifically, it becomes worse than the baselines according to either mAP or mAP$_{75:95}$. This indicates that a proper choice of $\alpha$ is crucial for $\alpha$-IoU losses. Our recommendation is to tune $\alpha \in [2, 3]$ for most applications or directly use $\alpha = 3$ when tuning is too expensive. Note that $\alpha \in [3, 4]$ may be a better choice when high levels of bbox regression accuracy is desired, e.g., mAP$_{75:95}$ is the preferred performance metric. It is possible that $\alpha < 1$ is a better choice for certain applications, although $\alpha$-IoU losses with $\alpha < 1$ perform consistently worse than the baselines in our experiments.

## 5 Conclusions

In this paper, we proposed a unified formula $\alpha$-IoU to generalize existing IoU-based losses to a new family of power IoU losses. By modulating the power parameter $\alpha$, $\alpha$-IoU offers the flexibility to achieve different levels of bbox regression accuracy when training an object detector. We analyzed the order preservingness and the loss/gradient reweighting properties of $\alpha$-IoU, and showed that $\alpha$-IoU can improve bbox regression accuracy through up-weighting the loss and gradient of high IoU objects. Experiments with multiple detection models and benchmark datasets demonstrated that $\alpha$-IoU losses can consistently outperform existing IoU-based losses, especially at the high Average Precisions (APs). $\alpha$-IoU has the potential to be widely applied in real-world object detection applications as 1) it improves existing IoU-based losses, 2) it benefits light models, 3) it is extremely advantageous on small datasets, and 4) it is more robust to noisy bboxes. For future work, we will explore new generalization formulas for other metric-derived loss functions [13], such as Dice, Hausdorff distance, and Chamfer distance losses.

## Societal Impacts

The proposed loss functions can help train high-performance object detectors for impactful applications such as self-driving, face recognition and video surveillance. While not our initial intention, these models could potentially be manipulated by adversaries or unauthorized users for malicious purposes. This could compromise the safety or privacy of certain individuals. We believe strict regulations should be established to prevent such illegitimate exploitations.

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
