# A  Property Analysis

Here, we provide detailed derivations and property analysis with proofs for Section 3.

In **Section 3.2**, we have $\lim_{\alpha\to 0}\mathcal{L}_{\alpha\text{-IoU}} = \mathcal{L}_{\log(\text{IoU})}$, where $\mathcal{L}_{\alpha\text{-IoU}} = \frac{1-IoU^\alpha}{\alpha}$, $\alpha > 0$ is the $\alpha$-IoU loss, and $\mathcal{L}_{\log(\text{IoU})} = -\log(IoU)$ is the log(IoU) loss.

*Proof.* From equation (1), and using L'Hôpital's rule,

$$\lim_{\alpha\to 0}\mathcal{L}_{\alpha\text{-IoU}} = \lim_{\alpha\to 0}\frac{1-IoU^\alpha}{\alpha} = \lim_{\alpha\to 0}\frac{\frac{d}{d\alpha}(1-IoU^\alpha)}{\frac{d}{d\alpha}\alpha}$$
$$= \lim_{\alpha\to 0} -IoU^\alpha\log(IoU) = -\log(IoU) = \mathcal{L}_{\log(\text{IoU})}. \tag{6}$$

$\square$

In **Section 3.3**, we only listed the properties that contribute the most to the superiority of $\mathcal{L}_{\alpha\text{-IoU}}$ over $\mathcal{L}_{\text{IoU}}$. The following properties are the complete list of properties with detailed derivations for $\mathcal{L}_{\alpha\text{-IoU}}$.

**Property 1** (Order Preservingness). *$\mathcal{L}_{\alpha\text{-IoU}}$ preserves the orders of IoU and $\mathcal{L}_{IoU}$: $IoU(B_i, B^{gt}) < IoU(B_j, B^{gt}) \iff \mathcal{L}_{IoU}(B_i, B^{gt}) > \mathcal{L}_{IoU}(B_j, B^{gt}) \iff \mathcal{L}_{\alpha\text{-IoU}}(B_i, B^{gt}) > \mathcal{L}_{\alpha\text{-IoU}}(B_j, B^{gt}).$*

When $IoU(B_i, B^{gt}) < IoU(B_j, B^{gt})$, we have,

$$\mathcal{L}_{\text{IoU}}(B_i, B^{gt}) = 1 - IoU(B_i, B^{gt}) > 1 - IoU(B_j, B^{gt}) = \mathcal{L}_{\text{IoU}}(B_j, B^{gt}),$$
$$\mathcal{L}_{\alpha\text{-IoU}}(B_i, B^{gt}) = 1 - IoU(B_i, B^{gt})^\alpha > 1 - IoU(B_j, B^{gt})^\alpha = \mathcal{L}_{\alpha\text{-IoU}}(B_j, B^{gt}). \tag{7}$$

$\mathcal{L}_{\alpha\text{-IoU}}$ strictly preserves the order of $\mathcal{L}_{\text{IoU}}$, thus $\arg\min_B \mathcal{L}_{\alpha\text{-IoU}}(B, B^{gt})$ is identical to $\arg\max_B IoU(B, B^{gt})$ and $\arg\min_B \mathcal{L}_{\text{IoU}}(B, B^{gt})$, i.e., the optimal solution (i.e., $\arg\max_B IoU(B, B^{gt})$) can be achieved by minimizing either $\mathcal{L}_{\alpha\text{-IoU}}$ or $\mathcal{L}_{\text{IoU}}$.

**Property 2** (Relative Loss Reweighting). *Compared with $\mathcal{L}_{IoU}$, $\mathcal{L}_{\alpha\text{-IoU}}$ adaptively reweights the relative loss of all objects by $w_{\mathcal{L}_r} = \mathcal{L}_{\alpha\text{-IoU}}/\mathcal{L}_{IoU} = 1+(IoU-IoU^\alpha)/(1-IoU)$, with $w_{\mathcal{L}_r}(IoU = 0) = 1$, and $\lim_{IoU\to 1} w_{\mathcal{L}_r} = \alpha$.*

The relative loss weight is,

$$w_{\mathcal{L}_r} = \mathcal{L}_{\alpha\text{-IoU}}/\mathcal{L}_{\text{IoU}} = (1 - IoU^\alpha)/(1 - IoU) = 1 + (IoU - IoU^\alpha)/(1 - IoU). \tag{8}$$

We will first prove the monotonicity of $g(x) = (x - x^\alpha)/(1 - x)$, which corresponds to the second term in $w_{\mathcal{L}_r}$. The conclusion is that $g(x) = (x - x^\alpha)/(1 - x)$, $x \in [0, 1)$ monotonically decreases w.r.t. $x$ when $0 < \alpha < 1$ and monotonically increases w.r.t. $x$ when $\alpha > 1$. Furthermore, we have $g(0) = 0$, and,

$$\lim_{x\to 1} g(x) = \lim_{x\to 1}\frac{x - x^\alpha}{1 - x} = \lim_{x\to 1}\frac{\frac{d}{dx}(x - x^\alpha)}{\frac{d}{dx}(1 - x)} = \lim_{x\to 1}\frac{1 - \alpha x^{\alpha-1}}{-1} = -1 + \alpha. \tag{9}$$

Therefore, $w_{\mathcal{L}_r}(IoU = 0) = 1$, and $\lim_{IoU\to 1} w_{\mathcal{L}_r} = \alpha$. When $IoU = 1$, $\mathcal{L}_{\alpha\text{-IoU}} = \mathcal{L}_{\text{IoU}} = 0$. In other words, when $0 < \alpha < 1$, $w_{\mathcal{L}_r}$ will decay from 1 to $\alpha$ monotonically with the increase of IoU, and when $\alpha > 1$, $w_{\mathcal{L}_r}$ will grow from 1 to $\alpha$ monotonically with the increase of IoU. For the bbox regression branch of object detection, the localization loss should up-weight the relative loss for high IoU objects (e.g., positive examples at $AP_{50}$) as the final performance is usually measured by $mAP_{50:95}$, where low IoU objects (negative examples at $AP_{50}$) are suppressed at evaluation. It basically means that low IoU objects contribute much less to the final performance evaluation, although some deviations between the predicted bboxes and their ground truth can be tolerated.

*Proof.* We have the first derivative of $g(x) = (x - x^\alpha)/(1 - x)$, $x \in [0, 1)$ w.r.t. $x$ as,

$$\frac{dg(x)}{dx} = \frac{\alpha x^\alpha - \alpha x^{\alpha-1} - x^\alpha + 1}{(1 - x)^2}, \tag{10}$$

where the denominator $(1 - x)^2$ is always positive. We thus can let $h(x)$ be the numerator of the first derivative function $\frac{dg(x)}{dx}$ as

$$h(x) = \alpha x^\alpha - \alpha x^{\alpha-1} - x^\alpha + 1, \tag{11}$$

then we have $\lim_{x \to 1} h(x) = 0$. Here, the second derivative of $g(x)$ is formulated as,

$$\frac{dh(x)}{dx} = \alpha^2 x^{\alpha-1} - \alpha(\alpha-1)x^{\alpha-2} - \alpha x^{\alpha-1} = \alpha(\alpha-1)x^{\alpha-2}(x-1), \tag{12}$$

where $\alpha x^{\alpha-2}(x-1) < 0$ since $\alpha > 0$, $x^{\alpha-2} > 0$, and $x - 1 < 0$. When $0 < \alpha < 1$, we have $\alpha - 1 < 0$. So $\frac{dh(x)}{dx} > 0$, i.e., $h(x)$ monotonically increases when $x \in [0, 1)$, which leads to $h(x) < 0$ because $h(1) = 0$. Therefore, $\frac{dg(x)}{dx} < 0$ and $g(x)$ monotonically decreases w.r.t. $x$ when $0 < \alpha < 1$. On the other hand, when $\alpha > 1$, we have $\alpha - 1 > 0$. So $\frac{dh(x)}{dx} < 0$, i.e., $h(x)$ monotonically decreases when $x \in [0, 1)$, which leads to $h(x) > 0$ because $h(1) = 0$. Therefore, $\frac{dg(x)}{dx} > 0$ and $g(x)$ monotonically increases w.r.t. $x$ when $\alpha > 1$. This completes the proof. $\square$

**Property 3** (Relative Gradient Reweighting). *Compared with $\mathcal{L}_{IoU}$, $\mathcal{L}_{\alpha\text{-}IoU}$ adaptively reweights the relative gradient of all objects by $w_{\nabla_r} = |\nabla_{IoU}\mathcal{L}_{\alpha\text{-}IoU}|/|\nabla_{IoU}\mathcal{L}_{IoU}| = \alpha IoU^{\alpha-1}$, with the turning point at $IoU = \alpha^{\frac{1}{1-\alpha}} \in (0, \frac{1}{e})$ when $0 < \alpha < 1$ and $IoU = \alpha^{\frac{1}{1-\alpha}} \in (\frac{1}{e}, 1)$ when $\alpha > 1$.*

The relative gradient weight is,

$$w_{\nabla_r} = \alpha IoU^{\alpha-1}, \tag{13}$$

which is a monotonically decreasing function when $0 < \alpha < 1$, and monotonically increasing when $\alpha > 1$, with $w_{\nabla_r}(IoU = \alpha^{\frac{1}{1-\alpha}}) = 1$, i.e., $|\nabla_{IoU}\mathcal{L}_{\alpha\text{-}IoU}| = |\nabla_{IoU}\mathcal{L}_{IoU}| = 1$. We prove that $\lim_{\alpha \to 1} \alpha^{\frac{1}{1-\alpha}} = \frac{1}{e}$ as follows:

$$\lim_{\alpha \to 1} \alpha^{\frac{1}{1-\alpha}} = \lim_{\alpha \to 1} e^{\frac{\log\alpha}{1-\alpha}} = e^{\lim_{\alpha \to 1} \frac{\log\alpha}{1-\alpha}} = e^{\lim_{\alpha \to 1} \frac{\frac{d}{d\alpha}\log\alpha}{\frac{d}{d\alpha}(1-\alpha)}} = e^{\lim_{\alpha \to 1} \frac{\frac{1}{\alpha}}{-1}} = \frac{1}{e}. \tag{14}$$

Hence, we can obtain that, when $0 < \alpha < 1$,

$$\begin{aligned} |\nabla_{IoU}\mathcal{L}_{\alpha\text{-}IoU}| &\geq |\nabla_{IoU}\mathcal{L}_{IoU}| \quad for \quad IoU \in [0, \alpha^{\frac{1}{1-\alpha}}], \\ |\nabla_{IoU}\mathcal{L}_{\alpha\text{-}IoU}| &< |\nabla_{IoU}\mathcal{L}_{IoU}| \quad for \quad IoU \in (\alpha^{\frac{1}{1-\alpha}}, 1], \end{aligned} \tag{15}$$

and when $\alpha > 1$,

$$\begin{aligned} |\nabla_{IoU}\mathcal{L}_{\alpha\text{-}IoU}| &\leq |\nabla_{IoU}\mathcal{L}_{IoU}| \quad for \quad IoU \in [0, \alpha^{\frac{1}{1-\alpha}}], \\ |\nabla_{IoU}\mathcal{L}_{\alpha\text{-}IoU}| &> |\nabla_{IoU}\mathcal{L}_{IoU}| \quad for \quad IoU \in (\alpha^{\frac{1}{1-\alpha}}, 1]. \end{aligned} \tag{16}$$

Therefore, $\mathcal{L}_{\alpha\text{-}IoU}$ up-/down-weights the relative gradient of low/high IoU objects when $0 < \alpha < 1$. In contrast, $\mathcal{L}_{\alpha\text{-}IoU}$ up-/down-weights the relative gradient of high/low IoU objects when $\alpha > 1$, which boosts the late training stage of a detector using high IoU objects. We empirically show that although $|\nabla_{IoU}\mathcal{L}_{\alpha\text{-}IoU}| \leq |\nabla_{IoU}\mathcal{L}_{IoU}|$ in $IoU \in [0, \alpha^{\frac{1}{1-\alpha}}]$ when $\alpha > 1$, there is no significant difference between $\mathcal{L}_{\alpha\text{-}IoU}$ and $\mathcal{L}_{IoU}$ in training a detector at the early training stage (Figure 3).

Apart from the above three properties, here we present two additional absolute properties of $\mathcal{L}_{\alpha\text{-}IoU}$.

**Property 4** (Absolute Loss Reweighting). *Compared with $\mathcal{L}_{IoU}$, $\mathcal{L}_{\alpha\text{-}IoU}$ adaptively reweights the absolute loss of all objects by $w_{\mathcal{L}_a} = \mathcal{L}_{\alpha\text{-}IoU} - \mathcal{L}_{IoU} = IoU - IoU^\alpha$, with the turning point at $IoU = \alpha^{\frac{1}{1-\alpha}} \in (0, \frac{1}{e})$ when $0 < \alpha < 1$ and $IoU = \alpha^{\frac{1}{1-\alpha}} \in (\frac{1}{e}, 1)$ when $\alpha > 1$.*

The absolute loss weight is

$$w_{\mathcal{L}_a} = IoU - IoU^\alpha, \tag{17}$$

which is non-positive when $0 < \alpha < 1$ and non-negative when $\alpha > 1$. From the first derivative of the above equation, we can obtain that the minimum/maximum (i.e., either negative or positive) absolute loss weight is achieved at $IoU = \alpha^{\frac{1}{1-\alpha}}$, which is the same as the IoU value for $w_{\nabla_r}(IoU = \alpha^{\frac{1}{1-\alpha}}) = 1$ in **Property 3**. For example, when $\alpha = 0.5$, $w_{\mathcal{L}_{a,\min}} = w_{\mathcal{L}_a}(IoU = 0.25) = -0.25$. Another example is when $\alpha = 2$, $w_{\mathcal{L}_{a,\max}} = w_{\mathcal{L}_a}(IoU = 0.5) = 0.25$. Therefore, $\mathcal{L}_{\alpha\text{-}IoU}$ is able to adjust $\mathcal{L}_{IoU}$ to be globally smaller (when $0 < \alpha < 1$) or larger (when $\alpha > 1$) than their original values by simply modulating the parameter $\alpha$. When $\alpha > 1$, this property creates more space for optimization for all levels of objects than the vanilla IoU.

**Property 5** (Absolute Gradient Reweighting). *Compared with $\mathcal{L}_{IoU}$, $\mathcal{L}_{\alpha\text{-}IoU}$ adaptively reweights the absolute gradient of all objects by $w_{\nabla_a} = |\nabla_{IoU}\mathcal{L}_{\alpha\text{-}IoU}| - |\nabla_{IoU}\mathcal{L}_{IoU}| = \alpha IoU^{\alpha-1} - 1$, with the turning point at $IoU = \alpha^{\frac{1}{1-\alpha}} \in (0, \frac{1}{e})$ when $0 < \alpha < 1$ and $IoU = \alpha^{\frac{1}{1-\alpha}} \in (\frac{1}{e}, 1)$ when $\alpha > 1$.*

The absolute gradient weight is,

$$w_{\nabla_a} = \alpha IoU^{\alpha-1} - 1. \tag{18}$$

$w_{\nabla_a} = 0$ is achieved at $IoU = \alpha^{\frac{1}{1-\alpha}}$, which is also the same IoU value as that in **Property 3** and **4**. Hence, we can also obtain that, when $0 < \alpha < 1$,

$$
\begin{aligned}
|\nabla_{\text{IoU}}\mathcal{L}_{\alpha\text{-IoU}}| &\geq |\nabla_{\text{IoU}}\mathcal{L}_{\text{IoU}}| \quad for \quad IoU \in [0, \alpha^{\frac{1}{1-\alpha}}], \\
|\nabla_{\text{IoU}}\mathcal{L}_{\alpha\text{-IoU}}| &< |\nabla_{\text{IoU}}\mathcal{L}_{\text{IoU}}| \quad for \quad IoU \in (\alpha^{\frac{1}{1-\alpha}}, 1],
\end{aligned}
\tag{19}
$$

and when $\alpha > 1$,

$$
\begin{aligned}
|\nabla_{\text{IoU}}\mathcal{L}_{\alpha\text{-IoU}}| &\leq |\nabla_{\text{IoU}}\mathcal{L}_{\text{IoU}}| \quad for \quad IoU \in [0, \alpha^{\frac{1}{1-\alpha}}], \\
|\nabla_{\text{IoU}}\mathcal{L}_{\alpha\text{-IoU}}| &> |\nabla_{\text{IoU}}\mathcal{L}_{\text{IoU}}| \quad for \quad IoU \in (\alpha^{\frac{1}{1-\alpha}}, 1].
\end{aligned}
\tag{20}
$$

This also indicates that, compared with $\mathcal{L}_{\text{IoU}}$, $\mathcal{L}_{\alpha\text{-IoU}}$ with $\alpha > 1$ up-weights the absolute gradient in $IoU \in (\alpha^{\frac{1}{1-\alpha}}, 1]$, which is the range of high IoU objects. Therefore, $\mathcal{L}_{\alpha\text{-IoU}}$ with $\alpha > 1$ will accelerate the learning of high IoU objects.

# B  Additional Experiments

## B.1  Detailed Training Setup

Here are the implementation details of all models used in this paper. We train YOLOv5, Faster R-CNN, and DETR using NVIDIA V100 GPUs.

**YOLOv5.** We train YOLOv5s and YOLOv5x with different losses following the original code's training protocol at `https://github.com/ultralytics/yolov5` with the released version being v4.0. We train both models from scratch using the same hyperparaemter in the file named "hyp.scratch.yaml". The configuration is set following the file "yolov5s.yaml" for YOLOv5s and "yolov5x.yaml" for YOLOv5x, respectively. The batch size is $64$, the initial learning rate is $0.01$, and the number of training epochs is $300$ in all experiments. The file "voc.yaml" is set for models trained on PASCAL VOC while the file "coco.yaml" is set for those trained on MS COCO.

**Faster R-CNN.** We train Faster R-CNN with different losses following the original code's training protocol at `https://github.com/open-mmlab/mmdetection/tree/master/configs/faster_rcnn`. The configuration is set following the file "faster_rcnn_r50_fpn.py" for Faster R-CNN with the backbone being ResNet-50-FPN. The file "schedule_1x.py" is set for models trained with 1x schedule and single scale. The checkpoint and logging configuration is set in "default_runtime.py". The batch size is $16$, the initial learning rate is $0.02$, and the number of training epochs is $12$ in all experiments. The file "coco_detection.py" is set for models trained on MS COCO. We do not train Faster R-CNN on PASCAL VOC.

**DETR.** We train DETR with different losses following the original code's training protocol at `https://github.com/open-mmlab/mmdetection/tree/master/configs/detr`. The configuration is set following the file "detr_r50_8x2_150e_coco.py" for DETR with the backbone being ResNet-50. "8x2" stands for using $8$ GPUs (we use NVIDIA V100 GPUs) in parallel with $2$ images trained on every GPU (i.e., batch size is $16$). The number of training epochs is $150$. The initial learning rate is $1e-4$ for the first $100$ epochs and $1e-5$ for the rest $50$ epochs. The checkpoint and logging configuration is set in "default_runtime.py". The file "coco_detection.py" is set for models trained on MS COCO. We also modify it for models trained on PASCAL VOC.

## B.2 More Results

We provide more experimental results here, including the sensitivity of $\mathcal{L}_{\alpha\text{-IoU}}$ to the second power parameter $\alpha_2$ for the penalty term, robustness of the detectors to small datasets trained by $\mathcal{L}_{\alpha\text{-IoU}}$, some failure cases, and visualizations of the synthesized noisy bboxes.

### B.2.1 Sensitivity to $\alpha_2$

In Section 3.2, we stated that $\mathcal{L}_{\alpha\text{-IoU}}$ is not sensitive to $\alpha_2$, and we reduced the two power parameters to a single parameter $\alpha$ by setting $\alpha_1 = \alpha_2$. Here, we empirically show the trivial difference among different $\alpha_2$ selections. 3-0.5, 3-1, 3-3 are used to denote $\alpha_1 = 3$ and $\alpha_2 = 0.5, 1, 3$ in equation (3), respectively. From Table 4, one can find that $\alpha$-IoU losses with different $\alpha_2$ values yield very close performances. Based on this observation, we choose to simply maintain $\alpha_1 = \alpha_2 = 3$ in all of our experiments.

Table 4: The performance of YOLOv5s trained using $\alpha$-IoU losses with different $\alpha$ values for the two loss terms. Results are obtained on the test set of PASCAL VOC 2007. mAP denotes $\text{mAP}_{50:95}$; $\text{mAP}_{75:95}$ denotes the mean AP over $\text{AP}_{75}, \text{AP}_{80}, \cdots, \text{AP}_{95}$.

| Loss | $\text{AP}_{50}$ | $\text{AP}_{55}$ | $\text{AP}_{60}$ | $\text{AP}_{65}$ | $\text{AP}_{70}$ | $\text{AP}_{75}$ | $\text{AP}_{80}$ | $\text{AP}_{85}$ | $\text{AP}_{90}$ | $\text{AP}_{95}$ | **mAP** | **$\text{mAP}_{75:95}$** |
|---|---|---|---|---|---|---|---|---|---|---|---|---|
| $\mathcal{L}_{\alpha\text{-GIoU}}$ 3-0.5 | 78.44 | 76.19 | 73.49 | 70.03 | 65.38 | 59.31 | 50.74 | 38.54 | 21.45 | 3.93 | 53.75 | 34.79 |
| $\mathcal{L}_{\alpha\text{-GIoU}}$ 3-1 | 77.94 | 75.83 | 73.19 | 69.80 | 65.21 | 58.25 | 49.84 | 37.68 | 21.54 | 3.68 | 53.29 | 34.20 |
| $\mathcal{L}_{\alpha\text{-GIoU}}$ 3-3 | 78.19 | 76.25 | 73.50 | 69.91 | 64.90 | 58.55 | 49.47 | 38.29 | 21.79 | 3.59 | 53.44 | 34.34 |
| $\mathcal{L}_{\alpha\text{-DIoU}}$ 3-0.5 | 77.33 | 75.68 | 73.21 | 69.57 | 65.27 | 58.42 | 49.75 | 38.42 | 21.23 | 3.74 | 53.26 | 34.31 |
| $\mathcal{L}_{\alpha\text{-DIoU}}$ 3-1 | 77.74 | 75.62 | 73.19 | 69.52 | 65.04 | 58.57 | 50.27 | 37.71 | 21.92 | 3.50 | 53.31 | 34.39 |
| $\mathcal{L}_{\alpha\text{-DIoU}}$ 3-3 | 78.42 | 76.13 | 73.57 | 70.16 | 65.83 | 59.14 | 50.20 | 38.31 | 21.71 | 3.50 | 53.70 | 34.57 |
| $\mathcal{L}_{\alpha\text{-CIoU}}$ 3-0.5 | 77.78 | 75.70 | 72.97 | 69.55 | 65.02 | 58.41 | 49.87 | 38.51 | 21.51 | 3.52 | 53.28 | 34.36 |
| $\mathcal{L}_{\alpha\text{-CIoU}}$ 3-1 | 78.02 | 75.85 | 73.18 | 69.78 | 65.28 | 58.34 | 50.11 | 37.85 | 21.68 | 3.66 | 53.37 | 34.33 |
| $\mathcal{L}_{\alpha\text{-CIoU}}$ 3-3 | 78.03 | 76.04 | 73.66 | 70.10 | 65.18 | 58.71 | 49.55 | 37.71 | 21.70 | 3.76 | 53.44 | 34.29 |

### B.2.2 Robustness to Small Datasets

This set of experiments simulate the real-world scenarios where the training data for a given task is very limited, e.g., only a few thousands of images. From the PASCAL VOC benchmark, we randomly select $50\%$ (containing $8,276$ images) and $25\%$ (containing $4,138$ images) trainval set 2007+2012, while maintaining the test set 2007 (containing $4,952$ images). As shown in Figure 7, $\alpha$-IoU is consistently more robust to different scales of the training set than the baseline losses. This superiority indicates that $\alpha$-IoU losses could be applied to practical application scenarios, where it is challenging to collect large amounts of data or annotations. In Table 5, we further observe that the relative improvement increases as the level of bbox regression accuracy rises across all losses and scales of the training set.

Table 5: Comparison results of YOLOv5s trained on small PASCAL VOC. Results are obtained on the test set of PASCAL VOC 2007. mAP denotes $\text{mAP}_{50:95}$; $\text{mAP}_{75:95}$ denotes the mean AP over $\text{AP}_{75}, \text{AP}_{80}, \cdots, \text{AP}_{95}$. "rela. improv." stands for the relative improvement. $\alpha = 3$ is used for all $\alpha$-IoU losses in all experiments.

| Loss | 50% PASCAL VOC | | | | | | 25% PASCAL VOC | | | | | |
|---|---|---|---|---|---|---|---|---|---|---|---|---|
| | $\text{AP}_{50}$ | $\text{AP}_{75}$ | $\text{AP}_{85}$ | $\text{AP}_{95}$ | **mAP** | **$\text{mAP}_{75:95}$** | $\text{AP}_{50}$ | $\text{AP}_{75}$ | $\text{AP}_{85}$ | $\text{AP}_{95}$ | **mAP** | **$\text{mAP}_{75:95}$** |
| $\mathcal{L}_{\text{IoU}}$ | 71.14 | 46.61 | 22.63 | 0.89 | 43.58 | 23.13 | 60.17 | 35.03 | 14.45 | 0.38 | 34.38 | 16.13 |
| $\mathcal{L}_{\alpha\text{-IoU}}$ | 71.01 | 47.88 | 27.48 | 1.24 | 44.81 | 25.62 | 59.96 | 36.02 | 17.02 | 0.60 | 34.93 | 17.57 |
| rela. improv. | -0.18% | 2.72% | 21.43% | 39.33% | 2.82% | 10.76% | -0.35% | 2.83% | 17.79% | 57.89% | 1.60% | 8.91% |
| $\mathcal{L}_{\text{DIoU}}$ | 71.29 | 46.95 | 23.79 | 0.9 | 44.02 | 23.52 | 59.92 | 34.54 | 14.06 | 0.25 | 33.95 | 15.71 |
| $\mathcal{L}_{\alpha\text{-DIoU}}$ | 70.93 | 48.09 | 27.41 | 1.56 | 44.83 | 25.76 | 59.68 | 35.90 | 16.72 | 0.51 | 34.69 | 17.42 |
| rela. improv. | -0.50% | 2.43% | 15.22% | 73.33% | 1.84% | 9.52% | -0.40% | 3.94% | 18.92% | 104% | 2.18% | 10.89% |
| $\mathcal{L}_{\text{GIoU}}$ | 70.93 | 47.14 | 24.57 | 0.80 | 43.89 | 23.95 | 59.75 | 33.88 | 13.90 | 0.38 | 33.8 | 15.56 |
| $\mathcal{L}_{\alpha\text{-GIoU}}$ | 71.19 | 48.37 | 26.92 | 1.67 | 45.10 | 25.82 | 59.58 | 35.74 | 16.96 | 0.59 | 34.88 | 17.23 |
| rela. improv. | 0.37% | 2.61% | 9.56% | 109% | 2.76% | 7.82% | -0.28% | 5.49% | 22.01% | 55.26% | 3.20% | 10.75% |

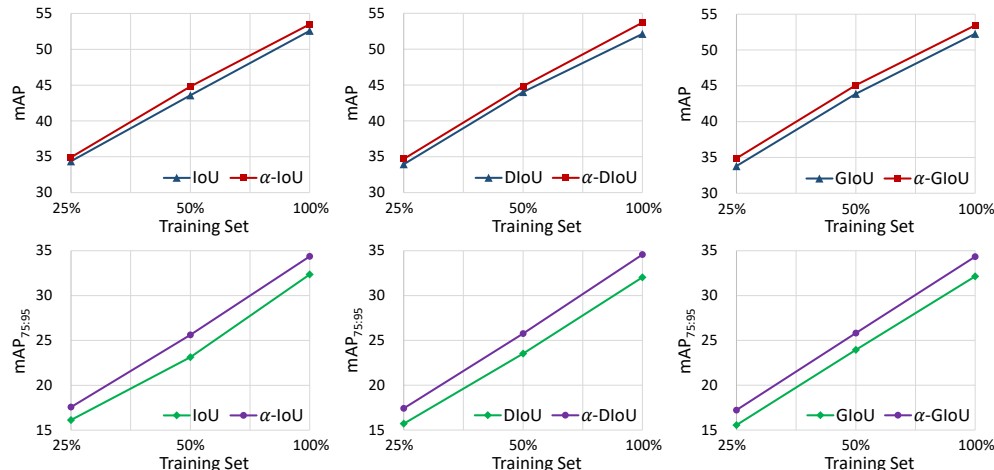

Figure 7: Comparison of the performance of YOLOv5s trained using different losses on different scales of datasets. Results are obtained on the test set of PASCAL VOC 2007. mAP denotes $\text{mAP}_{50:95}$; $\text{mAP}_{75:95}$ denotes the mean AP over $\text{AP}_{75}, \text{AP}_{80}, \cdots, \text{AP}_{95}$. $\alpha = 3$ is used for all $\alpha$-IoU losses in all experiments.

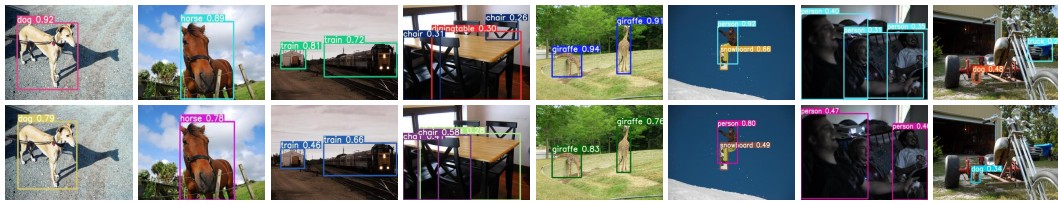

Figure 8: Example results on both the test set of PASCAL VOC 2007 (image $1$ to $4$) and the val set of MS COCO 2017 (image $5$ to $8$) using YOLOv5s trained by $\mathcal{L}_{\text{IoU}}$ (top row) and $\mathcal{L}_{\alpha\text{-IoU}}$ with $\alpha = 3$ (bottom row). $\mathcal{L}_{\alpha\text{-IoU}}$ may perform not as well as $\mathcal{L}_{\text{IoU}}$ in some cases.

## B.3 Failure Cases

We also find cases where $\mathcal{L}_{\alpha\text{-IoU}}$ performs not as well as $\mathcal{L}_{\text{IoU}}$ on both datasets (Figure 8). It is possible that $\mathcal{L}_{\alpha\text{-IoU}}$ classifies true positive objects with lower probability (image $1, 2, 5, 6$) since it focuses more on the localization branch than the classification branch by up-weighting the localization loss compared with the classification loss. However, the confidence threshold for classification (which is different from the IoU threshold for localization) is usually low. For example, $0.25$ is set for YOLOv5 at the inference stage, where we could still successfully detect most of these objects. Note that $\mathcal{L}_{\alpha\text{-IoU}}$ may misclassify objects (image $3$) or fail to detect them (image $4, 7, 8$) in images as well, although $\mathcal{L}_{\alpha\text{-IoU}}$ outperforms $\mathcal{L}_{\text{IoU}}$ in more cases generally.

## B.4 Examples of Noisy Bounding Boxes

Figure 9 visualizes some examples of the noisy bboxes synthesized in Section 4.3 under various noise rates $\eta$. Different colors represent different categories of objects.

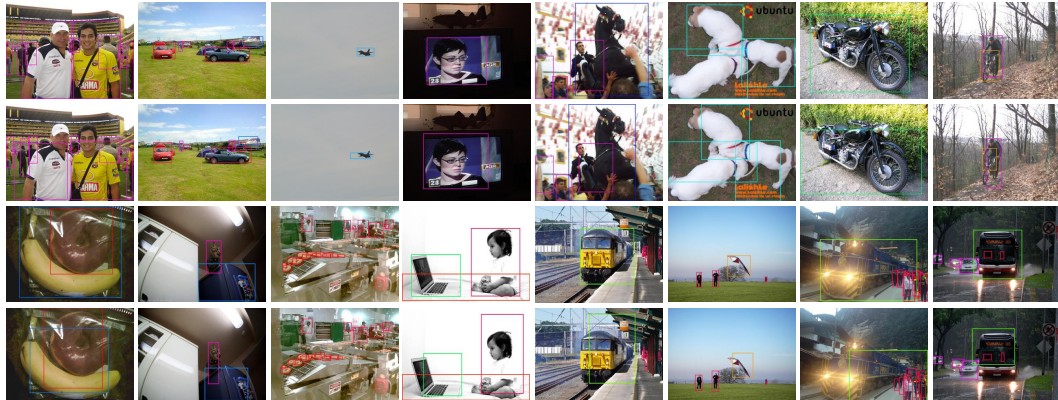

Figure 9: Examples of synthesized noisy bboxes on PASCAL VOC ($\eta = 0$ in the first row, $\eta = 0.3$ in the second row) and MS COCO ($\eta = 0$ in the third row, $\eta = 0.3$ in the fourth row).