# OpenReview forum: "$\alpha$-IoU: A Family of Power Intersection over Union Losses for Bounding Box Regression"
_NeurIPS.cc/2021/Conference — NeurIPS 2021 Poster_

### Official Review · Reviewer_TBNp · 2021-07-14

**Rating:** 6
**Confidence:** 2

**Summary:**

Bounding box regression is a fundamental task in computer vision. The most commonly used loss functions for box regression are the Intersection over Union (IoU) loss and its variants. This paper generalize existing IoU-based losses to a new family of power IoU losses that have a power IoU term and an additional power regularization term with a single power parameter $\alpha$. Experiments on multiple object detection benchmarks demonstrate the effectiveness  of the proposed loss.

**Ethics Review Area:**

["I don’t know"]

**Limitations And Societal Impact:**

1. Experiments should be designed with recent SOTA detectors, like FCOS, Faster R-CNN, Cascade R-CNN and ATSS.

**Main Review:**

1. Detectors should contain recent SOTA methods, like FCOS and ATSS.

2. Improvements is limited on YOLOv5 and DETR.


**Time Spent Reviewing:**

2 hours

---

> ### Author Response · Authors · 2021-08-10
> **Responses to Reviewer TBNp**
>
> Thanks for the thoughtful suggestion. Please find our response below.
>
> ---
> **Q1:** Detectors should contain recent SOTA methods, like FCOS, Faster R-CNN, Cascade R-CNN and ATSS.
>
> **A1:** The results of Faster R-CNN can be found in Table 2. Here, we have run an additional experiment with the suggested FCOS (R_50_FPN_1x) on the MS COCO dataset. The result can be found in the Table below. In the meanwhile, we would like to point out that YOLOv5 was released in 2020 (updated with several versions in 2021) which is more up-to-date and effective than Cascade R-CNN (2018) [1], FCOS (2019) [2], and ATSS (2020) [3], and is still one of the top performers on the SOTA leaderboard. We are more than happy to include more of the models (e.g., Cascade R-CNN and ATSS) to the revision. $\alpha=3$ is used here.
>
> (`FCOS (R_50_FPN_1x) on MS COCO`)
>
> |        Loss       |  mAP  |  AP50 |  AP75 |  APs  |  APm  |  APl  |
> |:-----------------:|:-----:|:-----:|:-----:|:-----:|:-----:|:-----:|
> |     $L_{IoU}$     | 38.10 | 56.70 | 41.40 | 22.60 | 41.60 | 50.40 |
> | $L_{\alpha -IoU}$ | 38.98 | 57.33 | 42.22 | 23.12 | 42.14 | 51.32 |
>
> ---
> **Q2:** Improvements is limited on YOLOv5 and DETR.
>
> **A2:** The improvement over YOLOv5x may not be significantly large. However, the improvement is certainly noticeable, i.e., ~0.6% in mAP and ~2.4% in mAP75:95 on average. In the meantime, we would like to draw the reviewer’s attention to the consistent improvements over YOLOv5s, DETR, and Faster-RCNN on both PASCAL VOC and MS COCO, under both clean and noisy bbox scenarios. Some of these improvements are noticeably large, for instance, on YOLOv5s in the noisy bbox scenario, up to 6.78% in mAP and 24.09% in mAP75:95 (see Table 3). We emphasize that all these improvements were achieved by adding a single power parameter. Moreover, we believe in a competitive area like object detection, even less than 1% is acceptable, for example, the improvement of ATSS (CVPR 2020) to FCOS (ICCV 2019) is less than 1% on average. More examples include GIoU (CVPR 2019) and DIoU/CIoU (AAAI 2020). Sometimes, even small improvements like a 1% mAP will still require a lot of architectural or algorithmic modifications, unlike our $\alpha$-IoU loss.
>
> ---
> [1] Cascade R-CNN: Delving into High Quality Object Detection, CVPR 2018.
>
> [2] FCOS: Fully Convolutional One-Stage Object Detection, ICCV 2019.
>
> [3] Bridging the Gap Between Anchor-based and Anchor-free Detection via Adaptive Training Sample Selection, CVPR, 2020.

---

> ### Author Response · Authors · 2021-08-31
> **Thanks to Reviewer TBNp**
>
> We would like to thank the reviewer for the valuable suggestions. We hope our response has adequately addressed your concerns regarding new experiments on FCOS and the improvement on YOLOv5 and DETR.  Kindly let us know if anything is unclear. We truly appreciate your valuable feedback that helps us improve our work.

---

> > ### Comment · Reviewer_TBNp · 2021-09-10
> > **leading to accept**
> >
> > After reading the author's response and other reviewers' comments, I would like to give accept.

---

> > > ### Author Response · Authors · 2021-09-13
> > > **Thanks to Reviewer TBNp for giving accept**
> > >
> > > We truly thank the reviewer for the valuable comments.

---

### Official Review · Reviewer_LZaD · 2021-07-15

**Rating:** 6
**Confidence:** 3

**Summary:**

The paper generalizes existing IoU based losses for bounding box regression into a family of alpha-IoU loss parametrized by one parameter. The paper shows improved object detection performance across multiple datasets and architectures by using different alpha values.

**Limitations And Societal Impact:**

Yes, the paper has addressed possible societal impacts.

**Main Review:**

Originality: the paper first proposes to generalize existing IoU losses and studies the effect of hyperparameters.  This leads to improved performance by choosing certain alpha values.

Clarity: the paper is well written and easy to follow

Quality: the paper conducts extensive experiments including noisy environment and sensitivity study.




**Time Spent Reviewing:**

0.5

---

> ### Author Response · Authors · 2021-08-10
> **Responses to Reviewer LZaD**
>
> Thanks for the helpful review. We are very happy to discuss further if the reviewer has more questions.

---

### Official Review · Reviewer_SBt3 · 2021-07-16

**Rating:** 7
**Confidence:** 4

**Summary:**

The paper proposes a generalization of the IoU-based losses which are widely used in applications
such as 2D object detection. This generalization enables the creation of a novel family of losses termed
alpha-IoU losses which: i) are demonstrated to have the ability to include widely adopted existing losses
e.g. Generalized IoU ii) give the possibility to create novel losses e.g. L_{\alpha-IoU}}=1-IoU^{\alpha} iii)
are easily tunable with a single hyper-parameter.
The authors propose quantitative comparisons wrt existing IoU losses and demonstrate the ability of
their proposed novel family to improve 2D detection performances especially at high-IoUs e.g. with the
AP75:95. The quantitative analyses have been performed i) on different datasets e.g. PASCAL VOC
and MS COCO; ii) with different models from high to low capacity e.g. YOLOv5 small and XL; iii) with
noisy annotations

**Limitations And Societal Impact:**

yes

**Main Review:**

Strengths:
- The paper proposes a generalization wrt widely used losses in the very popular topic
represented by 2D object detection
- This generalization is demonstrated to improve the performance of 2d detectors with the only
change of the loss function
- The novel losses are proven to be effective in noisy scenarios, which could be beneficial for
settings in which labels contains errors e.g. in semi-supervised or self-supervised settings
- The novel losses are proven to be effective with models of different sizes, and particularly
improving the performance of models with a lower capacity
- I believe this generalization to be novel and interesting

Weaknesses
- The proposed family of losses puts more weight on high IoU predictions rather than low IoU
ones. This is reflected in the gradients, as high IoU predictions will have “more gradient” than
low IoU ones. But is this a good thing? Why should we down-weight low IoU boxes if those are
exactly where we should improve the most? Focal loss does a great job in this sense, down-
weighting what is already classified well and therefore avoiding overfitting, while is it fair to say
that alpha-IoU losses encourage overfitting?
- I believe the authors put too much emphasis on the improvements in the AP75:95 metric but do
not adequately analyze the performances wrt other metrics e.g. AP50 or mAP.
- I believe the results shown on mAP to not adequately sustain the contribution but rather to raise
some questions e.g. why, according to the authors, does the AP50 and mAP remain basically
unaltered? Is the capability of the alpha-IoU limited to improving the IoU on “easier” boxes but
to be unable to improve the IoU overall and on “harder” boxes? Since mAP values are fair from
being saturated, does the alpha-IoU loss simply help the model to avoid learning “harder” boxes
and concentrate on the rest?

**Time Spent Reviewing:**

1

---

> ### Author Response · Authors · 2021-08-10
> **Responses to Reviewer SBt3**
>
> Thanks for the precise summarization and thoughtful comments. Please find our responses below.
>
> ---
> **Q1:** The proposed family of losses puts more weight on high IoU predictions rather than low IoU ones. This is reflected in the gradients, as high IoU predictions will have “more gradient” than low IoU ones. But is this a good thing? Why should we down-weight low IoU boxes if those are exactly where we should improve the most? Focal loss does a great job in this sense, down-weighting what is already classified well and therefore avoiding overfitting, while is it fair to say that alpha-IoU losses encourage overfitting?
>
> **A1:** The impact of $\alpha$-IoU on learning can be interpreted by combining both loss reweighting and gradient reweighting. Please allow us to clarify the following for $\alpha =3$.
>
> **$\alpha$-IoU up-weights IoU loss in general, but adaptively.** The left figure of Figure 1 shows that the green line ($\alpha =3$) is above the black line (the vanilla IoU loss), meaning $\alpha$-IoU loss will be higher than vanilla IoU everywhere except when IoU=0/1. This creates more space for optimization for all levels of objects in the **absolute** sense than vanilla IoU (see Property 4 (Absolute Loss Reweighting) in Appendix A), and the maximum boost occurs at IoU ~0.5, i.e., “boundary” objects. In the **relative** sense (Property 2 (Relative Loss Reweighting) at Lines 168-177), the up-weighting is adaptive to IoU with high IoU objects receiving more weight. This will allow the model to better exploit most learnable examples to boost learning.
>
> **Gradient up-weighting.** The gradient re-weighting scheme of $\alpha$-IoU creates a curriculum around the **turning point** that allows the model to learn at an adaptive **speed** on different IoU objects (See Property 3 (Relative Gradient Reweighting)). This is in contrast to the constant learning speed of the vanilla IoU loss for all objects (the blackline in the right plot of Figure 1). This can help the model focus more on what can be learned yet not fully learned objects (i.e., those with predicted IoUs above the turning point) instead of the objects that are too hard or even impossible to learn. Please also note that the gradient up-weighting is well-bounded by the power parameter $\alpha$ (see the right figure of Figure 1). For example, the maximum gradient weight is 3 and this only occurs when IoU reaches 1, i.e., $L_{IoU}$=0 (already converged).
>
> **Relation to Focal Loss and the overfitting issue.** Focal loss is a classification loss designed for the classification branch (see page 5 of the Focal loss paper) of object detectors. Focal loss dresses a special case of the class imbalance problem where there are a large number of the **easy** background class. By down-weighting the high confidence objects, Focal can avoid the easy class from overwhelming the **rare (hard)** foreground classes. We note that bbox regression is very different from classification. For bbox regression with diverse objects, it is very hard to achieve zero IoU loss for most of the objects. We believe this is also why focusing on high IoU objects can help bbox regression. As mentioned by Reviewer #3 pgM4, **“Cascade RCNN [1] also tries to gradually focus on higher IoU bboxes by increasing IoU thresholds”** to help train better detectors. At Lines 92-95, we also stated that both Rectified IoU (RIoU) [2] and Focal-EIoU [3] were designed to focus more on high IoU objects by increasing their gradients. In fact, $\alpha$-IoU can benefit more for small datasets (Appendix B.2.2), small models (Table 1) and noisy bboxes (Table 3), i.e., scenarios that are more prone to overfitting.
>
> We will add more interpretations (loss vs. gradient; absolute vs. relative) and understandings (adaptive reweighting and the learning dynamics) of our method along with empirical evidence to the revision.
>
> ---
> **Q2:** I believe the authors put too much emphasis on the improvements in the AP75:95 metric but do not adequately analyze the performances wrt other metrics e.g. AP50 or mAP.
>
> **A2:** While we have done an analysis of the AP50/mAP results, we agree that we could do more. Some of our analyses include “From Table 1, we can observe that $\alpha$-IoU losses surpass existing losses consistently across multiple models and datasets in terms of both mAP and mAP75:95, especially at the high bbox regression accuracy mAP75:95” at Lines 221-223, and “it is possible that $\alpha$-IoU losses may not perform well if measured by a single low AP metric. For example, there may be less than 0.5% performance drop at AP50, however, this is compensated by the significant boost at high APs” at Lines 232-234. While our method indeed performed extremely well at mAP75:95, we have also analyzed its performance using multiple evaluation metrics, including AP50, mAP, mAP75:95, and APs with other IoU thresholds. We can definitely add more analysis on the improvement of $\alpha$-IoU at the low APs based on these results.
>
> ---
> **Q3:** I believe the results shown on mAP to not adequately sustain the contribution but rather to raise some questions e.g. why, according to the authors, does the AP50 and mAP remain basically unaltered? Is the capability of the alpha-IoU limited to improving the IoU on “easier” boxes but to be unable to improve the IoU overall and on “harder” boxes? Since mAP values are fair from being saturated, does the alpha-IoU loss simply help the model to avoid learning “harder” boxes and concentrate on the rest?
>
> **A3:** While there is indeed concern around the hard examples, we may disagree with the reviewer that “the AP50 and mAP remain basically unaltered” nor “alpha-IoU loss simply avoids hard bboxes but only concentrates on the easy ones”. In fact, the performance in mAP has been greatly improved in most cases (e.g., Table 1 and 2), especially against noisy bboxes (Table 3). Arguably, noisy bboxes are hard rather than easy examples. We agree that low APs are not a strong point of  $\alpha$-IoU losses, which has been acknowledged at Lines 232-234. Please note that, in the three noisy bbox scenarios in Table 3 of Section 4.3, $\alpha$-IoU losses outperformed the baselines at AP50 and mAP by a large margin. We hope it is safe to say “$\alpha$-IoU does not ignore the hard bboxes”.
>
> ---
> [1] Cascade R-CNN: Delving into High Quality Object Detection, CVPR 2018.
>
> [2] Single-Shot Two-Pronged Detector with Rectified IoU Loss. ACM Multimedia, 2020.
>
> [3] Focal and Efficient IoU Loss for Accurate Bounding Box Regression, arxiv 2021.

---

> > ### Comment · Reviewer_SBt3 · 2021-09-13
> > **Comments after reading authors' rebuttal**
> >
> > Thanks to the authors for providing so detailed answers. I believe this paper will be an excellent contribution for the conference.

---

> > > ### Author Response · Authors · 2021-09-13
> > > **Thanks to Reviewer SBt3**
> > >
> > > We genuinely thank the reviewer for the helpful discussion.

---

### Official Review · Reviewer_Zi6N · 2021-07-17

**Rating:** 7
**Confidence:** 4

**Summary:**

This paper proposes a novel bounding box regression loss, $\alpha$-IoU loss, which is a family of power IoU loss. Essentially, the proposed loss is a power transformation of IoU loss. This paper demonstrates that applying such weighting improves detection performance in mAP, increases robustness to noisy label, and is especially powerful in small models.

**Limitations And Societal Impact:**

 - How does AP30-AP95 plot would look like where IoU threshold is on the x axis?
 - How does the proposed method impact precision / recall (not AP)?
 - How does this loss perform on difficult cases such as small or occluded objects?

I believe that answers to the questions above may better demonstrate the limitation of this work.

**Main Review:**

Overall, I have a very mixed feeling about this paper.

This paper is well-written. The paper thoroughly studies the proposed loss theoretically and empirically. The quantitative analysis clearly shows that the proposed method outperforms its baseline in mAP and is robust to noisy labels. I appreciate the huge effort this paper made to generalize all IoU variant losses into the family of power IoU loss. I see how this loss can be extremely helpful for training model on noisy label, since it can actively ignore bad labels that give low IoU.

At the same time, I have a huge concern about the general direction this paper is pursuing - focus more on easy tasks, do easy tasks better. I believe we should aim for the other way around.

This paper heavily reminds me of focal loss, a power transformation of cross entropy loss. While being the same power transformation, the goal of focal loss is to put less emphasis on easy classes and to focus more on hard / rare classes, which perfectly makes sense as a general goal of perception. In contrast, this paper focuses on optimizing a human-defined, arbitrary evaluation metric, reversing the actual task that we need to solve. Specifically, I would argue that AP95 is an unrealistic evaluation metric that even humans would totally fail. The proposed loss optimizes such an unrealistic evaluation metric, arguing that we should perfect easy tasks and focus less on hard tasks because our evaluation metric doesn’t care about hard tasks anyways. This makes me even question if mAP is the best evaluation metric for object detection. I have a concern that the proposed method may harm recall.


**Time Spent Reviewing:**

3hrs

---

> ### Author Response · Authors · 2021-08-10
> **Responses to Reviewer Zi6N**
>
> Thanks for your thoughtful comments. Please find our responses below to your concerns.
>
> ---
> **Q1:** I have a huge concern about the general direction this paper is pursuing - focus more on easy tasks, do easy tasks better. I believe we should aim for the other way around.
>
> **A1:** This is a fair concern. Please allow us to clarify the following for **$\alpha > 1$**.
>
> **Loss determines the amount (less vs. more) of learning.** When $\alpha > 1$, the left figure of Figure 1 shows that $\alpha$-IoU (e.g., the green line with $\alpha =3$) is above the vanilla IoU loss (i.e., the black line), meaning $\alpha$-IoU will be higher than vanilla IoU everywhere except when IoU=0/1. This creates more space for optimization for all levels of objects in the **absolute** sense than vanilla IoU (see Property 4 (Absolute Loss Reweighting) in Appendix A), and the maximum boost occurs at IoU ~0.5, i.e., “boundary” objects. In the **relative** sense (Property 2 (Relative Loss Reweighting) at Lines 168-177), the up-weighting is adaptive to IoU with high IoU objects receiving more weight. This will allow the model to better exploit most learnable examples to boost learning.
>
> **Gradient determines the speed (fast vs. slow) of learning.** The gradient reweighting scheme of $\alpha$-IoU creates a curriculum around the **turning point** that allows the model to learn at an adaptive **speed** on different IoU objects (See Property 3 (Relative Gradient Reweighting)). This is in contrast to the constant learning speed of the vanilla IoU loss for all objects (the blackline in the right plot of Figure 1). This can help the model focus more on what can be learned yet not fully learned objects (i.e., those with predicted IoUs above the turning point) instead of the objects that are too hard or even impossible to learn. Please also note that the gradient up-weighting is well-bounded by the power parameter $\alpha$ (see the right plot in Figure 1). For example, the maximum gradient up-weight is 3 when $\alpha=3$, and this only occurs when IoU reaches 1, i.e., $L_{IoU}$=0 (already converged).
>
> **Training is a dynamic process.** Easy examples will converge first towards IoU=1 (loss=0), while hard examples will be learned gradually and accelerated later on. Those examples that are almost **impossible-to-learn** tend to be ignored by $\alpha$-IoU losses. As mentioned by Reviewer pgM4, **“Cascade RCNN [1] also tries to gradually focus on higher IoU bboxes by increasing IoU thresholds”** to help train better detectors. At Lines 92-95, we also stated that both Rectified IoU (RIoU) [2] and Focal-EIoU [3] were designed to focus more on high IoU objects by increasing their gradients. We believe this is a unique characteristic of bbox regression with diverse difficulty levels of objects. We cannot agree more that, for classification tasks, focusing more on the hard examples generally improves performance. In bbox regression, it is hard to regress all bboxes to have 0 loss. As such, an adaptive strategy that helps the model exploit most learnable examples appears to be crucial.
>
> ---
> **Q2:** Specifically, I would argue that AP95 is an unrealistic evaluation metric that even humans would totally fail. The proposed loss optimizes such an unrealistic evaluation metric, arguing that we should perfect easy tasks and focus less on hard tasks because our evaluation metric doesn’t care about hard tasks anyways. This makes me even question if mAP is the best evaluation metric for object detection. I have a concern that the proposed method may harm recall.
>
> **A2:** Although AP95 is a very challenging evaluation metric, we included AP75-AP95 (and mAP75:95) to show that $\alpha$-IoU contributes mostly to the high AP results. The main results are still evaluated by mAP. The mAP metric has been well adopted by the community as it provides a (relatively) comprehensive evaluation across a wide range of APs. AP50 sets the minimum standard to ensure that the localization accuracy cannot be **bad**, i.e., the IoU between the predicted bboxes and their ground truth should be at least 0.5. This is understandable as the localization accuracy also impacts the classification results.
>
> Our method does not harm recall as shown in the following Table. It shows different Average Recall (AR) results of the YOLOv5s detectors trained by different methods on the MS COCO dataset. **AR: the maximum recall given a fixed number of detections per image, averaged over categories and IoUs.** Definitions of various Average Recalls in the Table below can be found in the official website of MS COCO [4]. $\alpha=3$ is used for all $\alpha$-IoU losses in all experiments.
>
> |       YOLOv5s      |    AR1    |    AR10   |   AR100   |    ARs    |    ARm    |    ARl    |
> |:------------------:|:---------:|:---------:|:---------:|:---------:|:---------:|:---------:|
> |      $L_{IoU}$     |   30.71   |    50.20   |   54.22   |   35.69   |   60.13   |    68.70   |
> |  $L_{\alpha -IoU}$ |   31.00   |   50.75   |   54.73   |   35.95   |   60.99   |   69.03   |
> |    rela. improv.   | **0.94%** | **1.10%** | **0.94%** | **0.73%** | **1.43%** | **0.48%** |
> |     $L_{DIoU}$     |   30.74   |   50.31   |   54.43   |   36.48   |   60.30   |   68.85   |
> | $L_{\alpha -DIoU}$ |   30.82   |   50.70   |   54.64   |   36.74   |   61.20   |   68.99   |
> |   rela. Improv.    | **0.26%** | **0.78%** | **0.39%** | **0.71%** | **1.49%** | **0.20%** |
>
> ---
> **Q3:** How does AP30-AP95 plot would look like where IoU threshold is on the x axis?
>
> **A3:** If AP is on the y axis and IoU threshold is on the x axis, they are decreasing curves with AP30 being the highest point (maybe at the left of the figure) and AP95 being the lowest point (maybe at the right of the figure) for every case. Curves with better performance tend to reach to the upper right region. We guess the reviewer may want to see some visualization for Table 3 and more. This is a good suggestion of visualization and we will add these figures in the Appendix.
>
> ---
> **Q4:** How does the proposed method impact precision / recall (not AP)?
>
> **A4:** As shown in the above Table, our method can also improve ARs (various Average Recalls). APs are just different average **precisions** over all categories of objects under specific IoU thresholds. The AP and mAP results reported in the main paper indicate that our method can improve precisions. We will add a discussion on the precision and recall to the appendix.
>
> ---
> **Q5:** How does this loss perform on difficult cases such as small or occluded objects?
>
> **A5:** From the results of APs in the $9^{th}$ column of Table 2 (here APs denotes AP for small objects, with definitions in [4]), we could see that our method consistently surpasses the baselines. In addition, from the new results of ARs (AR for small objects, with definitions in [4]) in the $4^{th}$ column of the above Table, improvement is also noticeable. We are more than happy to add the results on occluded objects to the revision.
>
> ---
> [1] Cascade R-CNN: Delving into High Quality Object Detection, CVPR 2018.
>
> [2] Single-Shot Two-Pronged Detector with Rectified IoU Loss. ACM Multimedia, 2020.
>
> [3] Focal and Efficient IoU Loss for Accurate Bounding Box Regression, arxiv 2021.
>
> [4] https://cocodataset.org/#detection-eval

---

> > ### Comment · Reviewer_Zi6N · 2021-09-01
> > **Thank you for clarification**
> >
> > I thank the authors of the paper for huge efforts they made to lift all of my concerns. As I wrote in my original review, I have no doubt that this paper is well written. AR authors reported quantitative relieved my concerns and additional explanation qualitatively explains that my concern was invalid. Authors made huge effort addressing my concerns and I am convinced that this paper would be a great contribution to the conference.

---

> > > ### Author Response · Authors · 2021-09-01
> > > **Thanks to Reviewer Zi6N**
> > >
> > > We truly thank the reviewer for the valuable discussion. Your insightful comments, especially Q1, greatly help us better explain our intuitions and motivations. We will add core parts of the discussion to the revision.

---

> ### Author Response · Authors · 2021-08-31
> **Thanks to Reviewer Zi6N**
>
> We would like to thank the reviewer for taking the time to review our paper and the valuable feedback, and in particular for recognizing the theoretical and empirical aspects of our proposed generalization.
>
> We hope our response has adequately addressed your concerns regarding our approach "focus more on easy tasks, do easy tasks better" and the adoption of AP95 as one of the evaluation metrics. In addition, we hope we have clarified the performance on difficult cases such as small or occluded objects.
>
> Kindly let us know if anything is unclear. We truly appreciate your valuable feedback and comments that help us further highlight/clarify the important parts of our work.

---

### Official Review · Reviewer_pgM4 · 2021-07-17

**Rating:** 7
**Confidence:** 4

**Summary:**

This paper proposes a very simple extension to the previous IoU losses for object detection bounding regression, by introducing a power term and a power parameter $\alpha$. The paper gives both theoretical analyses and experimental analyses to the proposed approach. Results on PASCAL VOC and COCO show that the proposed approach outperforms other IoU losses consistently.

**Limitations And Societal Impact:**

Yes.

**Main Review:**

- $Originality:$ This paper proposes an $\alpha$-loss which introduces a power term and a power parameter $\alpha$. The proposed approach is interesting.

- $Quality:$ The submission is technically sound. The claims proposed in this paper are well supported by theoretical and experimental analyses.

- $Clarity:$ The paper is well written and easy to follow.

- $Significance:$ This paper shows consistent improvements comparing to different IoU loss baselines. Apart from performance improvements on standard benchmarks, the paper also shows the robustness of the proposed approach on small scale dataset and noisy data. Importantly, the simple extension makes it easier to be adopted by others. There are only a few questions about the experiments. 1) The DETR paper shows improvements by combining IoU loss with L1 loss. It would be interesting to show that the proposed $\alpha$-loss could also benefit from such combination. 2) It would be interesting to show results of applying the proposed approch to stronger detectors, such Deformable-DETR and Faster/Mask/Cascade-RCNN with stronger CNN backbones. 3) The proposed $\alpha$-loss has an interesting characteristic that it could let the detector focusing on higher IoU boxes by using a larger $\alpha$. Cascade RCNN tries to gradually focus on higher IoU boxes by increasing IoU thresholds. It would be interesting to show that the proposed approach could help Cascade RCNN like detectors by gradually increasing $\alpha$ instead of increasing IoU thresholds. (This could be a future work.)

In summary, this is an interesting paper with thorough theoretical and experimental analyses. Therefore, I would like to give an accept to this paper.

**Time Spent Reviewing:**

3

---

> ### Author Response · Authors · 2021-08-10
> **Responses to Reviewer pgM4**
>
> Thank you very much for the insightful comments. Please find our responses below to your questions.
>
> ---
> **Q1:** The DETR paper shows improvements by combining IoU loss with L1 loss. It would be interesting to show that the proposed α-loss could also benefit from such a combination.
>
> **A1:** Thanks for the suggestion. We will add the suggested experiment to the Appendix.
>
> ---
> **Q2:** It would be interesting to show results of applying the proposed approach to stronger detectors, such as Deformable-DETR and Faster/Mask/Cascade-RCNN with stronger CNN backbones.
>
> **A2:** Thanks for the suggestion. We have conducted an additional experiment with the suggested Faster R-CNN with a stronger backbone (ResNet-101-FPN) on MS COCO. Please find the new results below. $\alpha=3$ was used.
>
> (`Faster R-CNN (ResNet-101-FPN) on MS COCO`)
>
> |        Loss       |  mAP  |  AP50 |  AP75 |  APs  |  APm  |  APl  |
> |:-----------------:|:-----:|:-----:|:-----:|:-----:|:-----:|:-----:|
> |     $L_{IoU}$     | 41.80 | 62.50 | 45.80 | 24.70 | 46.20 | 53.70 |
> | $L_{\alpha -IoU}$ | 42.58 | 63.38 | 46.33 | 25.65 | 47.06 | 54.60 |
>
> ---
> **Q3:** The proposed α-loss has an interesting characteristic that it could let the detector focus on higher IoU boxes by using a larger α. Cascade RCNN tries to gradually focus on higher IoU boxes by increasing IoU thresholds. It would be interesting to show that the proposed approach could help Cascade RCNN like detectors by gradually increasing α instead of increasing IoU thresholds. (This could be a future work.)
>
> **A3:** Thanks for the suggestion. We are definitely interested to see if a simple power parameter can achieve a similar effect in Cascade RCNN. We will add this experiment to our revision.
>
> ---

---

> > ### Comment · Reviewer_pgM4 · 2021-09-02
> > **Keep my original rating**
> >
> > After reading other reviews and authors' feedbacks, I would like to keep my original accept rating.

---

> > > ### Author Response · Authors · 2021-09-02
> > > **Thanks to Reviewer pgM4**
> > >
> > > We genuinely thank the reviewer for the valuable comments.

---

### Official Review · Reviewer_k9Mu · 2021-07-19

**Rating:** 7
**Confidence:** 4

**Summary:**

This paper extends the existing IoU loss to a new form $\alpha$-IoU, which helps to improve bounding box regression in learning object detectors. Besides, this loss is also more adaptive and robust to noisy labels. Experiments using various detectors on major detection benchmarks have validated the claims.


**Ethical Concerns:**

I have read the Ethics Guidelines, and I don't find any concerns from this paper.
1. Potential negative societal impacts: It doesn't involve any social experiments, and doesn't use human-derived data.
2. General ethical conduct: the dataset used in this work are public ones.

**Limitations And Societal Impact:**

1. Limitations: Sec.4.4 has discussed the $\alpha$ issue and provided some insights about how to set it. I also agree that this is arguably the biggest limitation of this work.
2. Societal Impacts are discussed in the main paper last section, which is reasonable and thoughtful.

**Main Review:**

Overall, I think this is a great work whose strengths can be summarized as:
1. The proposed loss formulation is relatively simple and easy to implement. One can easily swap any forms of IoU loss (eg., vanilla IoU, DIoU, GioU, etc.) with the proposed version.
2. The experimental section is thorough. Both one-stage and two-stage models are tested, and different IoU losses are compared and ablated. The robustness section is interesting.
3. Sec 3.3 and the corresponding contents in Appendix are very informative and useful.
4. Writing is mostly clear and related works are properly studied and compared.

Weakness
1. My biggest concern about this loss ($\alpha>1$) is: when the training just gets started and loss value is high, the gradient value is small.  However, when the it converges and loss is small, the gradient becomes big. This would cause two bad things: (1) slow down training, and (2) introduce instability to the convergence state.
2. In practice, It's hard to find the optimal $\alpha$ for a new dataset or model. Also, it's not clear to me when should we use $\alpha<1$ or $\alpha>1$. It would be interesting to see more in-depth results/discussions on this. Theoretical analysis would be better. In fact, I don't even know whether $\alpha=3$ is dataset specific or something else.
3. Even though the robustness experiment is interesting, it lacks reasonable explanation or analysis. I'm not very convinced about why the proposed form is more robust. What makes it particularly robust against this kind of noise (inaccurate b-boxes).

Questions:
How important is it to introduce $\alpha$ into both the IoU loss and the penalty terms. Which one is more important for these two-term losses?

Misc
1. I find Fig.3 hard to read. The performance difference is too small on this figure, making the lines stay too close. I would recommend making it simpler and easy to read.
2. Line44  "nor decreasing the inference speed is confusing. During inference, most of time we don't compute any loss. Maybe replace with "training speed"?

I'm looking forward to the discussion. Please correct my if I understand anything wrong. I'm willing to increase my rating if my doubts are resolved.

**Time Spent Reviewing:**

5

---

> ### Author Response · Authors · 2021-08-10
> **Responses to Reviewer k9Mu**
>
> Thanks for your accurate summarization of our work and all those insightful comments. Please find our responses below.
>
> ---
> **Q1:** My biggest concern about this loss (α>1) is: when the training just gets started and the loss value is high, the gradient value is small. However, when it converges and loss is small, the gradient becomes big. This would cause two bad things: (1) slow down training, and (2) introduce instability to the convergence state.
>
> **A1:** This is a fair concern.
>
> **Early training stage:** As shown in Figure 2, $\alpha$-IoU does not slow down the convergence though. Our conjecture is that tuning down the gradients of low-IoU (high loss) examples in the early stage (e.g., first 100 epochs) may have a smoothing effect (reducing the high variance in parameter update caused by hard examples) that helps stabilize (or at least not significantly slow down) the model training when gradients are large in the early stage.
>
> **Near to convergence:** As also shown in Figure 2, $\alpha$-IoU losses are able to boost the later training stage (e.g., after 200 epochs) through up-weighting the losses and gradients of high IoU objects. This is because, in this stage, up-weighting the gradients of high IoU samples is relatively safe, as the loss is extremely low for these examples, so is their original gradients. And the learning rate in this stage is also very small. We would also like to point out that, for $\alpha=3$, the up-weighting is well-bounded, i.e., only by a factor of $\leq 3$ relative to the vanilla IoU loss (see the right plot in Figure 1). The up-weighting is 3 for IoU=1 objects, which in this case, will have 0 training loss thus exerting no impact on training.
>
> ---
> **Q2:** In practice, It's hard to find the optimal α for a new dataset or model. Also, it's not clear to me when we should use α<1or α>1. It would be interesting to see more in-depth results/discussions on this. Theoretical analysis would be better. In fact, I don't even know whether α=3 is dataset specific or something else.
>
> **A2:** We found that $\alpha \in [2,3]$ is very competitive and $\alpha=3$ often performs the best across all tested models, datasets, and noisy/clean bbox scenarios. As we analyzed in Property 3 (Relative Gradient Reweighting), $\alpha$ defines the turning point of the up-weighting scheme and the optimal turning point should be ~0.5 if mAP above AP50 is the primary concern. And this gives us $\alpha > 2$. This property is both data-agnostic and model-agnostic. So we conjecture that $\alpha=3$ or $\alpha \in [2,3]$ could generalize well to other models and datasets, or is at least a good start for $\alpha$ search.
>
> Therefore, a general rule of thumb is to use $\alpha>1$ if region proposals with high localization accuracy are of main interest. Otherwise, if high localization accuracy is not essential or low APs are equally important, then a $\alpha<1$ can be used. In practice, as the mAP above AP50 is often treated as the minimum standard, we would recommend $\alpha \in[2,3]$ in most cases.
>
> Thanks for the helpful discussion. We will definitely improve our explanation in the revision.
>
> ---
> **Q3:** Even though the robustness experiment is interesting, it lacks reasonable explanation or analysis. I'm not very convinced about why the proposed form is more robust. What makes it particularly robust against this kind of noise (inaccurate b-boxes).
>
> **A3:** Noisy bbox examples are often the hard examples that are of low IoUs and high losses. By focusing more on high IoU (less on low IoU) objects, $\alpha$-IoU losses can actively suppress the learning of the extremely noisy bbox examples, thus mitigating the overfitting of the detector to the noisy bboxes and enhancing robustness of the model. We believe this is an intrinsic and nice property of $\alpha$-IoU losses. However, we are still looking for a proper definition of the symmetric (robustness) condition [1] required for the theoretical analysis in bbox regression. So, we chose to show the robustness of $\alpha$-IoU losses empirically in Section 4.3. We will leave the theoretical justification to future work.
>
> ---
> **Q4:** How important is it to introduce α into both the IoU loss and the penalty terms. Which one is more important for these two-term losses?
>
> **A4:** Introducing $\alpha$ to the IoU term is more important than to the penalty term. This can be inferred from the results in Table 1. For example, $L_{\alpha -IoU}$ is adding $\alpha$ to the IoU loss, while $L_{\alpha -DIoU}$ is further adding $\alpha$ to the penalty term, since $L_{DIoU}=L_{IoU} + P$, with $P$ being the central point distance as its penalty term. We will add an ablation analysis in the Appendix to address this.
>
> ---
> **Q5:** I find Fig.3 hard to read. The performance difference is too small on this figure, making the lines stay too close. I would recommend making it simpler and easy to read.
>
> **A5:** Thanks for pointing this out. We will separate Figure 3 into four separate plots to improve readability.
>
> ---
> **Q6:** Line44 "nor decreasing the inference speed is confusing. During inference, most of the time we don't compute any loss. Maybe replace it with "training speed"?
>
> **A6:** Thanks for pointing this out. We will fix this in the next version.
>
> ---
> [1] Robust loss functions under label noise for deep neural networks, AAAI 2017.

---

> > ### Comment · Reviewer_k9Mu · 2021-08-23
> > **Some further feedback & comments**
> >
> > Thanks for answering my question in details. Most of my doubts have been answered.
> >
> > From my prospective, some discussions here are quite interesting (Q1, Q2). They might either be informative to the future users of this loss, or inspire some follow-up works. Given the fact that our discussions here are mainly empirical or originated from conjecture, I would recommend the authors to include these stuff into the limitation or discussion section of the paper.
> >
> > Moreover, there are still lots of space for improving the writing. Fixing them would increase the reability greatly.
> >
> > PS. raised my rating to 7. Good paper.

---

> > > ### Author Response · Authors · 2021-08-23
> > > **Thanks to Reviewer k9Mu**
> > >
> > > Thanks again for your further feedback. We will definitely include the discussions in our revision and improve the writing part as suggested.

---

### Official Review · Reviewer_BMDv · 2021-07-21

**Rating:** 7
**Confidence:** 4

**Summary:**

This paper proposes a unified loss formulation for a series of IoU losses. This paper explores its theoretical properties, and conduct several experiments to demonstrate the proposed \alpha-IoU's effectiveness.

**Limitations And Societal Impact:**

Yes

**Main Review:**

Overall I think this is a good paper. I only have the following minor things for improvement.

1. Plotting the curves of \alpha-IoU v.s. \alpha for some IoU values. This could help the choice of \alpha
2. In Fig. 3, the legend for different approaches are redundant. For example, the \alpha-IoU w/o noise appears twice. You can remove one of them.

**Time Spent Reviewing:**

3 hours

---

> ### Author Response · Authors · 2021-08-10
> **Responses to Reviewer BMDv**
>
> Thanks for your constructive suggestions. We will plot the curves in the Appendix and change Figure 3 as suggested. More suggestions are welcome.

---

> > ### Comment · Reviewer_BMDv · 2021-08-25
> > **Comments about the rebuttal**
> >
> > After checking the authors' responses and other reviewers' comments, I decide to keep my original rating as "7: Good paper, accept"

---

> > > ### Author Response · Authors · 2021-08-25
> > > **Thanks to Reviewer BMDv**
> > >
> > > Thanks again for your review and time. We really appreciate that.

---

### Decision · Program_Chairs · 2021-09-28

**Decision:**

Accept (Poster)

**Comment:**

Reviewers agreed that this is a solid paper that deserves acceptance. Authors are highly encouraged to address the key comments reported by reviewers as well as to implement all the improvements (as indicated by authors in the rebuttal) in the final camera-ready version.

**Consistency Experiment:**

NeurIPS has a long history of experimentation. In 2014, NeurIPS ran an experiment in which 10% of submissions were reviewed by two independent committees to quantify the randomness in the review process. This year, we repeated a variant of this experiment to see how the quality of the review process has changed over time.  This paper was part of the experiment and was therefore assigned to two committees (consisting of reviewers, an Area Chair, and a Senior Area Chair) that reached independent decisions.  If both committees made the same recommendation, this recommendation was followed. If a single committee recommended acceptance, the paper was accepted (with the exception of a few cases in which the other committee identified what we considered a fatal flaw, e.g., an error in a key result).

This copy’s committee reached the following decision: **Accept (Poster)**

The other committee assigned to the paper recommended **Reject**.  You can find the other set of reviews, along with any follow up discussion with the authors here:
https://openreview.net/forum?id=h3qQzodaAq7